# 🔥FIRE: A Dataset for Feedback Integration and Refinement Evaluation of Multimodal Models

**Pengxiang Li**[1,2*] **Zhi Gao**[2,3*] **Bofei Zhang**[2*] **Tao Yuan**[2] **Yuwei Wu**[1,5✉]
**Mehrtash Harandi**[4] **Yunde Jia**[5,1] **Song-Chun Zhu**[2,3,6] **Qing Li**[2✉]

[1]Beijing Key Laboratory of Intelligent Information Technology,
School of Computer Science & Technology, Beijing Institute of Technology
[2]State Key Laboratory of General Artificial Intelligence, BIGAI
[3]State Key Laboratory of General Artificial Intelligence, Peking University
[4] Department of Electrical and Computer System Engineering, Monash University
[5]Guangdong Laboratory of Machine Perception and Intelligent Computing, Shenzhen MSU-BIT University
[6]Department of Automation, Tsinghua University
mm-fire.github.io

Figure 1: The comparison of the **feedback-refining** capability among different models. While the original LLaVA hardly improves its responses, our model trained on FIRE can effectively integrate the user feedback and produce much better responses, which are closer to those of GPT-4V.

## Abstract

Vision language models (VLMs) have achieved impressive progress in diverse applications, becoming a prevalent research direction. In this paper, we build FIRE, a feedback-refinement dataset, consisting of 1.1M multi-turn conversations that are derived from 27 source datasets, empowering VLMs to spontaneously refine their responses based on user feedback across diverse tasks. To scale up the data collection, FIRE is collected in two components: FIRE-100K and FIRE-1M, where FIRE-100K is generated by GPT-4V, and FIRE-1M is freely generated via models trained on FIRE-100K. Then, we build FIRE-Bench, a benchmark to comprehensively evaluate the feedback-refining capability of VLMs, which contains 11K feedback-refinement conversations as the test data, two evaluation settings, and a model to provide feedback for VLMs. We develop the FIRE-LLaVA model by fine-tuning LLaVA on FIRE-100K and FIRE-1M, which shows remarkable feedback-refining capability on FIRE-Bench and outperforms untrained VLMs by 50%, making more efficient user-agent interactions and underscoring the significance of the FIRE dataset.

---

*Equal contribution.  ✉ Corresponding author.

38th Conference on Neural Information Processing Systems (NeurIPS 2024) Track on Datasets and Benchmarks.

# 1    Introduction

Vision language models (VLMs), such as LLaVA [45], GPT-4V [55], and Gemini [63], have shown impressive instruction-following abilities across various tasks [76, 43, 11, 15, 75, 33] by integrating large language models (LLMs) [65, 23] with visual encoders [14, 58]. However, VLMs may sometimes produce undesirable outputs, possibly due to omitting important details in images or misunderstanding the instructions, which prompts the need for the **feedback-refining** capability beyond the normal instruction-following ability. This capability enables VLMs to spontaneously refine their responses based on user feedback, as depicted in Fig. 1, enhancing the efficiency and smoothness of interactions between users and visual assistants.

In this paper, we build FIRE, a dataset for Feedback Integration and Refinement Evaluation of VLMs. FIRE is composed of 1.1M high-quality multi-turn feedback-refinement conversations, derived from 27 source datasets across a wide range of tasks, such as visual question answering [18], image captioning [8], OCR reasoning [54, 60], document understanding [20], math reasoning [47], and chart analysis [51]. To scale up the data collection, FIRE is collected in two stages. In the first stage, we randomly sample ∼100K image-instruction-response triplets from data sources. We use each triplet to instruct GPT-4V to simulate a dialogue between a student and a teacher: the student answers the question and the teacher provides feedback to help the student improve its answer. We filter out generated low-quality conversations, such as those with too many turns or no improvement, rendering 100K high-quality feedback-refinement conversations, named FIRE-100K. In the second stage, we fine-tune two LLaVA-NeXT [44] models on FIRE-100K: one is trained as a student to refine its answer with the feedback, and the other is trained as a teacher to generate feedback for the student's answer. We simulate dialogues between the student and the teacher models using ∼1M data points from the data sources, rending a split named FIRE-1M. In this case, the full FIRE dataset consists of 1.1M feedback-refinement conversations in two splits FIRE-100K and FIRE-1M.

To comprehensively evaluate the feedback-refining capability of VLMs, we build FIRE-Bench that has 11K feedback-refinement conversations derived from 16 source datasets, including test splits from 8 seen datasets in FIRE-100K and FIRE-1M, as well as 8 unseen datasets from recently-proposed popular multimodal benchmarks. Using FIRE-Bench, we design two evaluation settings: fixed dialogues and free dialogues. In fixed dialogues, we compare the model's refined response with ground truth in the generated conversations in FIRE-Bench, given a fixed dialogue history. In free dialogues, we let the model freely interact with a teacher model about instructions in FIRE-Bench, and test how fast & how much the model can improve its answers based on the feedback provided by the teacher model.

We develop FIRE-LLaVA by fine-tuning LLaVA-NeXT [44] on FIRE-100K and FIRE-1M. The evaluation results on FIRE-Bench shows that FIRE-LLaVA exhibits significant improvements based on feedback in conversations, exceeding the original LLaVA-NeXT model by $50\%$. These results underscore the significance of FIRE-100K and FIRE-1M in enhancing feedback integration, while FIRE-Bench provides an evaluation platform to analyze refinements. We expect that FIRE could motivate future exploration of the feedback-refining capability of VLMs.

In summary, our contributions are three-fold. (1) We introduce FIRE, a dataset containing 1.1M feedback-refinement conversations across a wide range of tasks, where 100K data is generated by GPT-4V and 1M data is freely generated by simulating dialogues between tuned open-source models. (2) We introduce the FIRE-Bench benchmark, composed of 11K conversations and a teacher model, providing comprehensive evaluations for the feedback-refining capability in two settings: fixed dialogues and free dialogues. (3) We develop FIRE-LLaVA, an advanced VLM that could improve its responses based on feedback, making efficient interaction between users and VLMs.

# 2    Related Work

## 2.1    Vision Language Models

Building open-source VLMs to compete with closed-source models like GPT-4V [55] and Gemini [63] is a hot research topic. BLIP [36, 35] and Flamingo [1] are pioneering models that combine LLMs with visual encoders to enhance cross-modal understanding and reasoning abilities. LLaVA [45], InstructBLIP [13], MMICL [73], and MiniGPT4 [76] develop the instruction tuning ability of VLMs by introducing a large number of instruction-response pairs. Along this way, some work focuses on the visual grounding or editing ability of VLMs [7], such as Kosmos-2 [57], SearchVLMs [34], MINI-GPTv2 [6], Qwen-VL [3], and UltraEdit [74], improving the region understanding for VLMs.

InternVL [11] and mini-Gemini [38] develop powerful visual encoders for high-resolution images, and CuMo adopts a mixture-of-experts (MOE) architecture to manage diverse data better. Compared with existing VLMs, our FIRE-LLaVA has a more powerful feedback-refining capability across diverse tasks, which can spontaneously refine responses based on user feedback, leading to efficient and smooth interaction with users.

## 2.2 Vision-Language Data Generation

Recent attention has increasingly focused on synthesizing vision-language data. The ShareGPT4V dataset [8] leverages GPT-4V to generate 1.2M image-text pairs with detailed descriptions, making better alignments. LLaVA-Instruct-150K [45] is a general visual instruction tuning dataset constructed by feeding captions and bounding boxes to GPT-4. After that, many efforts have been made to enhance the data diversity of instruction tuning data. LLaVAR [71], MIMIC-IT [31], and SVIT [72] further scale up it to 422K, 2.8M, and 4.2M, respectively. InternLM-XComposer [69] produces interleaved instruction and image data, enabling advanced image-text comprehension and composition. Mini-Gemini [38] and ALLaVA [4] use GPT-4V to exploit visual information and generate high-quality instruction data. LRV-Instruction [42] creates positive and negative instructions for the hallucinating inconsistent issue. A recent work DRESS [10] collects 66K feedback data and trains VLMs for the feedback-refining capability. Unlike DRESS, which only uses data from LLaVA-Instruct-150K, our feedback-refinement data is from richer sources (27 datasets) across more tasks (math reasoning, chart understanding, and OCR *etc.*). Moreover, FIRE has significantly more data than DRESS (1.1M *vs.* 66K), where 1M data is freely produced via dialogues of student and teacher models, leading to significant data expansion but a similar cost of data generation.

## 2.3 Feedback Learning in Multimodal Models

Learning from feedback is a promising research direction, playing an important role in human-robot interaction [39, 12]. Existing feedback learning methods can be roughly divided into two categories: planned feedback learning and impromptu feedback learning. Planned feedback learning updates models based on user feedback, and thus can generalize to new data but cannot provide refined responses immediately. CLOVA [17] and Clarify [30] are representative methods that automatically collect data to learn new knowledge. LLaVA-RLHF [62] collects human preference and trains VLMs via reinforcement learning. Self-refine[50] shows that LLMs could improve their responses by iteratively refining their outputs based on self-generated feedback. Impromptu feedback learning can immediately refine responses but have less generalization since they usually do not update models, which is widely studied in LLMs [2, 37, 64]. Liao *et al.* [40] use VLMs themselves as verifiers that produce feedback to correct recognition results. VolCaNo [29] generates data specifically for refinement to address visual hallucinations. DRESS [10] generates helpfulness, honesty, and harmlessness responses via impromptu feedback learning. Different from DRESS, we improve the correctness and details of responses via impromptu feedback learning across diverse tasks.

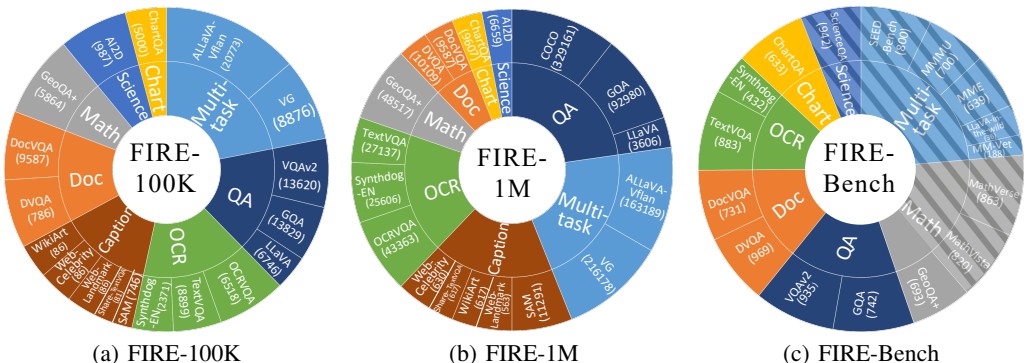

(a) FIRE-100K          (b) FIRE-1M          (c) FIRE-Bench

Figure 2: Data sources in FIRE. Shaded are new data sources in FIRE-Bench.

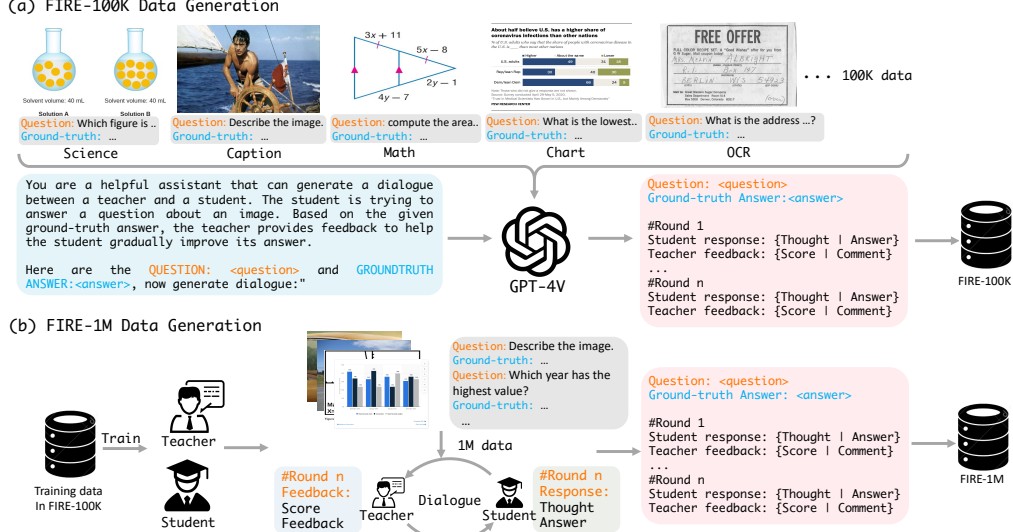

Figure 3: The pipeline to create FIRE-100K and FIRE-1M data.

## 3  Feedback Integration and Refinement Evaluation (FIRE)

This section presents the FIRE dataset, outlining its task definition, data collection methodology for FIRE-100K and FIRE-1M, and the creation of FIRE-Bench. Finally, we provide an analysis of FIRE.

### 3.1  Task Definition

**Data Source.** To enhance the diversity and comprehensiveness of our dataset, we compile more than 1.1M image-instruction-response triples from 27 source datasets (more details can be found in Appendix B), being used to generate FIRE-100K, FIRE-1M, and FIRE-Bench, as shown in Fig. 2. These datasets cover tasks including visual question answering, image captioning, complex reasoning, OCR, chart/table/document analysis, math problems, science question answering *etc*.

**Data format.** We formulate our data as $\{I, q, gt, \{r_i, f_i\}_{i=1}^n\}$, where $I$ denotes the image, $q$ is the instruction, $gt$ is the ground truth answer, and $\{r_i, f_i\}_{i=1}^n$ corresponds to the conversations in $n$ turns. In the $i$-th turn, $r_i$ is the response from VLMs, composed of the thought (how to refine the response based on feedback) and a new answer; $f_i$ is the feedback, involving a score $a_i$ (0-10) for the response $r_i$ and textual comments.

### 3.2  FIRE-100K

We feed images, instructions, ground truth answers from 18 datasets, and a designed textual prompt to GPT-4V that generates high-quality feedback-refinement conversations in a one-go manner, as shown in Fig. 3 (a). We ask GPT-4V to play two roles: a student and a teacher, and generate a conversation between the two roles, where the student's responses are improved by incorporating feedback from the teacher. After generation, we filter out low-quality conversations with no score improvements or more than 6 turns, since we expect that VLMs could learn to quickly and efficiently improve their responses from our data. Finally, we obtain 100K conversations, shown in Fig. 2(a).

### 3.3  FIRE-1M

We use FIRE-100K to fine-tune LLaVA-NeXT [44] and obtain two models: FIRE100K-LLaVA and FD-LLaVA, which are used to act as the student and the teacher, respectively (training details are shown in Sec. 4). We sample 1M data from 18 source datasets and generate feedback-refinement conversations via the following steps, as shown in Fig. 3 (b). (1) We feed an image and instruction to the student that generates a response. (2) We feed the image, instruction, ground truth answer, and the response to the teacher that generates feedback. If the score $a$ in the feedback $a \geq 8$ or the number of turns exceeds 3, we stop the conversation; otherwise, we go to step (3). (3) We feed the feedback to the student that generates a refined response and go back to step (2). Finally, we obtain 1M data, shown in Fig. 2(a)

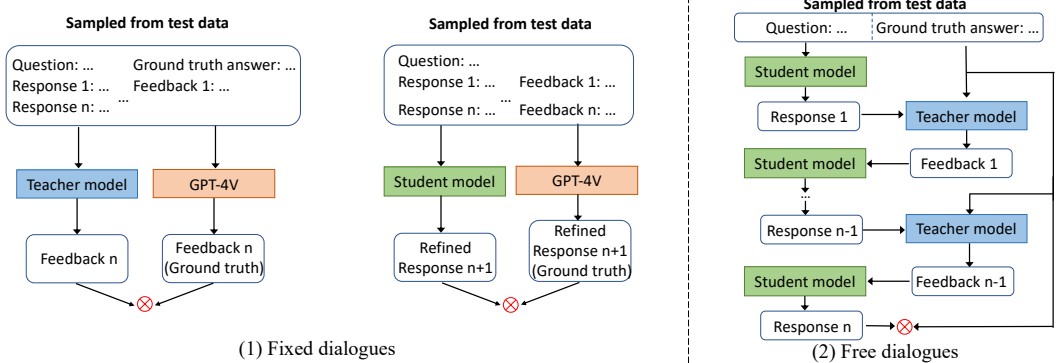

Figure 4: We use two settings to evaluate student and teacher models.

### 3.4 FIRE-Bench

To comprehensively evaluate the feedback-refining ability of VLMs, we introduce FIRE-Bench, containing 11K high-quality feedback-refinement conversations. As shown in Fig. 2(c), FIRE-Bench is derived from 16 source datasets, including 8 seen datasets (test splits) from FIRE-100K and FIRE-1M, as well as 8 new datasets from recently-proposed popular multimodal benchmarks, which is used to evaluate the generalization of the feedback-refining ability across different types of tasks. Similar to FIRE-100K, we sample 11K examples from the data sources and prompt GPT-4V to generate the feedback-refinement conversations.

#### 3.4.1 Evaluation Settings

We design two evaluation settings: fixed dialogues and free dialogues to evaluate the performance of the student and teacher models, as shown in Fig. 4.

**Fixed Dialogues.** In fixed dialogues, we evaluate whether the student and teacher models can generate appropriate responses and feedback given the conversation history, and their performance is evaluated by being compared with GPT-4V generated feedback and response, using the BLEU [56] and CIDEr [66] metrics to measure the textual alignment. For the predicted score $\hat{a}_i$ in feedback, we regard the score $a_i$ generated by GPT-4V as the ground truth and adopt *mean absolute error (MAE)*: $MAE = \frac{1}{K} \sum_{k=1}^{K} |a_k - \hat{a}_k|$, where there are $K$ test data totally. The teacher model may fail to follow instructions and does not generate a score in feedback for some cases. Here, we simply set $|a_i - \hat{a}_i| = 10$ for these cases.

**Free Dialogues.** We use a student model and a teacher model to perform free dialogues and evaluate how fast and how much the student model can improve its answers based on the feedback from the teacher model. The stopping condition for dialogues is that the obtained scores from the teacher model do not increase or exceed a pre-defined threshold (we set $8$ in experiments).

We introduce four metrics: average turn (AT), average dialogue refinement (ADR), average turn refinement (ATR), and refinement ratio (RR) for free dialogues.

(1) *Average Turn (AT)*. The AT metric evaluates how fast a VLM could achieve a satisfactory result based on feedback. We measure the number of turns $n_k$ in the conversation to solve the $k$-th data, where VLMs refine their responses until the obtained score exceeds the pre-defined threshold. We set a punishment number as $p = 10$, the maximum number of turns as $n_{max} = 5$. If VLMs fail to obtain a satisfactory score in $n_{max}$ turns, then $n_k = p$. For clearer comparisons with the baseline model (*e.g.*, the original LLaVA-NeXT model), we normalize it according to the AT of the baseline model,

$$AT = \frac{1}{K} \sum_{k=1}^{K} n_k / T_{baseline}, \tag{1}$$

where $T_{baseline}$ is the average turn of the baseline model. A smaller value of AT means better performance.

(2) *Average Dialogue Refinement (ADR)*. The ADR metric evaluates how much knowledge VLMs could learn from feedback in a dialogue. In solving the $k$-th data, we use $a_{k,1}$ to denote the obtained score for the initial response and use $a_{k,n_k}$ to denote the obtained score for the response in the final turn. ADR averages the score improvements of each conversation as

$$ADR = \frac{1}{K} \sum_{k=1}^{K} a_{k,n_k} - a_{k,1}. \tag{2}$$

A larger value of ADR means better performance.

(3) *Average Turn Refinement (ATR)*. ATR evaluates how much knowledge VLMs could learn from feedback in one turn. ATR averages the score improvements in each turn of $K$ samples as

$$ATR = \frac{1}{K} \sum_{k=1}^{K} \frac{1}{n_k - 1} (a_{k,n_k} - a_{k,1}). \tag{3}$$

A larger value of ATR means better performance.

(4) *Refinement Ratio (RR)*. RR measures the proportion of data that have a wrong initial response and a correct final response (*i.e.*, how much data are corrected based on feedback), computed by

$$RR = \frac{1}{K} \sum_{k=1}^{K} \mathbb{1}_{a_{k,n_k} \geq 8} - \mathbb{1}_{a_{k,1} \geq 8}, \tag{4}$$

where $\mathbb{1}_{a_{k,n_k} \geq 8}$ means if $a_{k,n_k} \geq 8$, $\mathbb{1}_{a_{k,n_k} \geq 8} = 1$, and 0 otherwise. A larger value of RR means better performance. Note that, for the $k$-th sample, if $n_k = 1$, we remove it from the K samples to compute AT, ADR, ATR, and RR.

## 3.5 Dataset Analysis

We provide three key statistics: score, turn, and length, for the collected feedback-refinement conversations. **Score.** We show the distribution of initial scores in Fig. 5(a), which reflects the starting state of the conversation. They mainly fall in the interval $[3, 8]$, showing that FIRE covers diverse starting states of conversations. Improved scores per turn are shown in Fig. 5(b), which reflects the learning effect. It ranges from $[2, 8]$, similar to actual situations, where high improvements are obtained in easy cases and small improvements are obtained in hard cases, showcasing the diversity of data. Improved scores per dialogue are shown in Fig. 5(c), and the improvements in most cases are 5-7, demonstrating the data quality of FIRE, where most data have obvious improvements, helping VLMs to efficiently learn to improve their responses. The score distributions of FIRE-100K, FIRE-1M, and FIRE Bench are not completely consistent, making the data more diverse. **Turn.** The turn distribution of conversations is shown in Fig. 5(d). Most conversations have 2-4 turns, indicating an efficient and concise feedback process. This measure suggests that most conversations reach a satisfactory level of refinements. A small number of turns in FIRE informs VLMs to perform effective dialogues. **Length.** The length distributions of responses and feedback are shown in Fig. 5(e) and Fig. 5(f), respectively. Most responses or feedback are less than 100 words. It shows concise dialogues in FIRE, aligning with real-world scenarios where users typically engage in brief exchanges rather than lengthy discussions.

# 4 Model

Our model architecture has the same design as LLaVA-NeXT-8B [43] that uses CLIP [58] as a frozen image encoder with a two-layer multi-layer perceptron vision-language connector. For the LLM part, we use the same architecture as the LLaMA3-8B [53]. We use LLaVA-NeXT-8B to initialize the VLMs and use LoRA to fine-tune the LLaVA-NeXT-8B for a student model and a teacher model.

## 4.1 Student Model

Given an $n$-turn conversation $\{I, q, gt, \{r_i, f_i\}_{i=1}^{n}\}$, we train a student model to fit responses $r_i$ for $i \geq 2$ using the cross-entropy loss,

$$\min \mathbb{E}_{(I,q,gt,\{r_i,f_i\}_{i=1}^{n}) \sim \mathbb{D}} \left[ -\sum_{i=2}^{n} \log P(r_i | I, q, \{r_j, f_j\}_{j=1}^{j=i-1}) \right], \tag{5}$$

where $\mathbb{D}$ is the used dataset. We first use FIRE-100K as $\mathbb{D}$ to train a student model FIRE100K-LLaVA, then use all training data (FIRE-100K and FIRE-1M) to train a final student model FIRE-LLaVA.

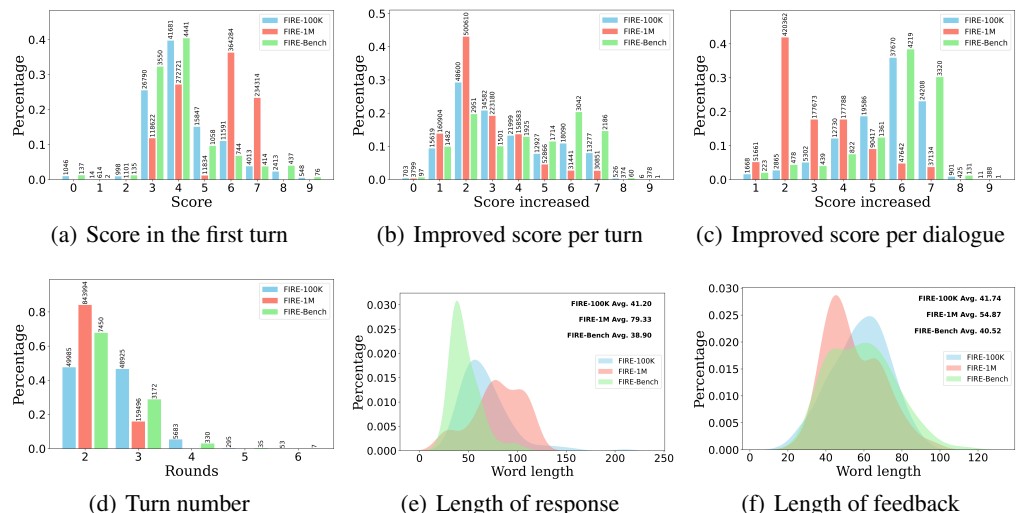

| (a) Score in the first turn | (b) Improved score per turn | (c) Improved score per dialogue |
|---|---|---|
| (d) Turn number | (e) Length of response | (f) Length of feedback |

Figure 5: Data statistics on FIRE-100K, FIRE-1M, FIRE-Bench.

Table 1: Comparisons between LLaVA-NeXT-8B and FIRE100K-LLaVA on 10 benchmarks. Benchmark names are abbreviated for space limits. GQA [22]; VQAv2 [18];VizWiz [19]; TextVQA [61]; SQA$^I$:ScienceQA-IMG [48]; LLaVA$^W$: LLaVA-Bench-in-the-wild [45];MMB: MMBench [46]; MME$^P$: MME Perception [16]; MME$^C$: MME Cognition [16]; MM-Vet [67].

| Method | GQA | VQAv2 | VizWiz | TextVQA | SQA$^I$ | LLaVA$^W$ | MMB | MME$^P$ | MME$^C$ | MM-Vet |
|---|---|---|---|---|---|---|---|---|---|---|
| LLaVA-NeXT-8B | **65.9** | 79.0 | 52.0 | **69.8** | **77.3** | 78.5 | 74.4 | **1546.0** | **331.4** | 44.9 |
| FIRE-LLaVA | 65.8 | **82.9** | **59.8** | 68.4 | 76.8 | **81.5** | **78.5** | 1534.8 | 321.1 | **45.3** |

## 4.2 Teacher Model

Given a $n$-turn conversation $\{I, q, gt, \{r_i, f_i\}_{i=1}^n\}$, we train a teacher model to fit the feedback $f_i$ for $i \geq 1$ using the cross-entropy loss,

$$\min \mathbb{E}_{(I,q,gt,\{r_i,f_i\}_{i=1}^n)\sim\mathbb{D}} \left[ -\sum_{i=1}^n \log P(f_i|I, q, gt, \{r_j, f_j\}_{j=1}^{j=i-1}, r_i) \right], \quad (6)$$

where we use FIRE-100K as $\mathbb{D}$ and obtain the teacher model FD-LLaVA.

## 5 Experiments

We conduct experiments to evaluate both the student and teacher models trained on FIRE. We first provide experimental details and then comprehensively evaluate models in multiple settings.

### 5.1 Experimental Details

**Training Data.** To avoid the catastrophic forgetting issue, we combine the training data in FIRE with the LLaVA-665K [45] (released by Open-LLaVA-1M [9]) to train the student and teacher models.

**Training Details.** In the training process of both the student and teacher models, we freeze the image encoder and the image-language connector, and fine-tune the language decoder using LoRA [21]. In the implementation of LoRA, we set the rank as 64 and only apply LoRA on the query and key projection matrices in all attention layers of the language decoder. This setting only involves 0.4% parameters of LLaMA3-8B. We use the AdamW optimizer, where a cosine annealing scheduler is employed, the learning rate is $2e-4$, the batch size is 64, and we train 1 epoch over all data. The training process for a student (or teacher) model requires about 128 A100-80GB GPU hours.

Table 2: Results of the student model in fixed dialogues.

| Model | BLEU-1 (↑) | BLEU-2 (↑) | BLEU-3 (↑) | BLEU-4 (↑) | CIDEr (↑) |
|---|---|---|---|---|---|
| LLaVA-NeXT-8B | 0.33 | 0.23 | 0.17 | 0.13 | 0.60 |
| FIRE-LLaVA | **0.54** | **0.46** | **0.39** | **0.34** | **2.36** |

Table 3: Results of the teacher model in fixed dialogues.

| Model | BLEU-1 ($\uparrow$) | BLEU-2 ($\uparrow$) | BLEU-3 ($\uparrow$) | BLEU-4 ($\uparrow$) | CIDEr ($\uparrow$) | MAE ($\downarrow$) |
|---|---|---|---|---|---|---|
| LLaVA-NeXT-8B | 0.34 | 0.21 | 0.15 | 0.10 | 0.51 | 1.88 |
| FD-LLaVA | **0.55** | **0.45** | **0.39** | **0.33** | **2.27** | **0.30** |

Table 4: Results in free dialogues overall test data in FIRE.

| Model | AT ($\downarrow$) | ADR ($\uparrow$) | ATR ($\uparrow$) | RR ($\uparrow$) |
|---|---|---|---|---|
| LLaVA-NeXT-8B | 1 | 0.97 | 0.41 | 0.25 |
| FIRE100K-LLaVA-8B | 0.92 | 1.27 | 0.55 | 0.34 |
| FIRE-LLaVA-8B | **0.84** | **1.56** | **0.66** | **0.39** |

## 5.2 Evaluation in Instruction Following

Considering that fine-tuning VLMs may encounter the catastrophic forgetting problem, we evaluate the instruction-following ability of FIRE-LLaVA, using 10 commonly used multimodal benchmarks, as shown in Tab. 1. Our model achieves comparable performance to the original LLaVA-NeXT-8B model, showing that we do not compromise the instruction-following ability when learning the feedback-refining ability.

## 5.3 Evaluation in Fixed Dialogues

We evaluate the performance of FIRE-LLaVA, and FD-LLaVA in fixed dialogues. The evaluation of FIRE-LLaVA is shown in Tab. 2, where we report the results of BLEU-1, BLEU-2, BLEU-3, BLEU-4, and CIDEr. The performance of FD-LLaVA is shown in Tab. 3, where we report the results of BLEU-1, BLEU-2, BLEU-3, BLEU-4, CIDEr, and MAE. We observe that using FIRE, FIRE-LLaVA and FD-LLaVA generates good responses and feedback, having better performance than the original LLaVA-NeXT-8B model on all metrics. FIRE-LLaVA could well refine the responses, like GPT-4V. FD-LLaVA can generate more accurate feedback, including comments (see BLEU and CIDEr) and scores (see MAE), demonstrating the effectiveness of our teacher model FD-LLaVA that can discover undesirable responses.

## 5.4 Evaluation in Free Dialogues

We employ a student model and a teacher model to perform free dialogues. We evaluate LLaVA-NeXT-8B, FIRE100K-LLaVA, and FIRE-LLaVA as the student model, and use FD-LLaVA to act as the teacher model. We report the average turn (AT), average dialogue refinement (ADR), average turn refinement (ATR), and refinement ratio (RR) on FIRE-Bench. Results are shown in Tab. 4. We observe that a LLaVA model trained on FIRE has improved feedback-refining ability. On the ADR, ATR, and RR metrics, FIRE-LLaVA achieves more than 50% improvements by LLaVA-NeXT, making an efficient user-agent interaction. Meanwhile, adding FIRE-1M to training data has better performance than only using FIRE-100K, showing the data quality of FIRE-1M.

We also show the detailed results on 8 seen source datasets and 8 new source datasets, as shown in Tab. 5 and Tab. 6, respectively. Our models achieve improvements on both seen and new datasets, showing the generalization of feedback-refining ability across different types of data and tasks.

Table 5: Detailed test results (AT ($\downarrow$), ADR ($\uparrow$), ATR ($\uparrow$), and RR ($\uparrow$)) on 8 seen source datasets.

| Model | VQAv2 | | | | GQA | | | | TextVQA | | | | ChartQA | | | |
|---|---|---|---|---|---|---|---|---|---|---|---|---|---|---|---|---|
| | AT | ADR | ATR | RR | AT | ADR | ATR | RR | AT | ADR | ATR | RR | AT | ADR | ATR | RR |
| LLaVA-NeXT | 1.00 | 1.45 | 0.42 | 0.40 | 1.00 | 1.51 | 0.51 | 0.43 | 1.00 | 0.91 | 0.34 | 0.26 | 1.00 | 0.71 | 0.39 | 0.25 |
| FIRE100K-LLaVA | 0.86 | 1.83 | 0.55 | 0.54 | 0.81 | 1.93 | 0.63 | 0.58 | 0.95 | 1.20 | 0.49 | 0.33 | 1.07 | 1.03 | **0.56** | 0.27 |
| FIRE-LLaVA | **0.78** | **2.08** | **0.59** | **0.56** | **0.81** | **2.06** | **0.70** | **0.58** | **0.77** | **1.51** | **0.56** | **0.42** | **0.79** | **1.15** | 0.53 | **0.36** |

| Model | DocVQA | | | | DVQA | | | | GEOQA+ | | | | Synthdog | | | |
|---|---|---|---|---|---|---|---|---|---|---|---|---|---|---|---|---|
| | AT | ADR | ATR | RR | AT | ADR | ATR | RR | AT | ADR | ATR | RR | AT | ADR | ATR | RR |
| LLaVA-NeXT | 1.00 | 0.97 | 0.56 | 0.24 | 1.00 | 1.66 | **0.50** | 0.42 | 1.00 | 0.14 | 0.07 | 0.08 | 1.00 | 0.14 | 0.05 | 0.04 |
| FIRE100K-LLaVA | 1.06 | 0.84 | 0.51 | 0.22 | 0.79 | 1.87 | 0.46 | **0.51** | **0.84** | 0.70 | 0.33 | **0.28** | **0.93** | 0.18 | 0.07 | **0.08** |
| FIRE-LLaVA | **0.81** | **1.65** | **0.97** | **0.41** | **0.74** | **1.97** | 0.46 | 0.50 | **0.84** | **0.74** | **0.35** | 0.27 | 0.95 | **0.19** | **0.08** | 0.06 |

Table 6: Detailed test results (AT (↓), ADR (↑), ATR (↑), and RR (↑)) on 8 new source datasets.

| Model | MathVista | | | | MathVerse | | | | MMMU | | | | MME | | | |
|---|---|---|---|---|---|---|---|---|---|---|---|---|---|---|---|---|
| | AT | ADR | ATR | RR | AT | ADR | ATR | RR | AT | ADR | ATR | RR | AT | ADR | ATR | RR |
| LLaVA-NeXT | 1.00 | 0.84 | 0.45 | 0.19 | 1.00 | 0.14 | 0.13 | 0.08 | 1.00 | 0.94 | 0.53 | 0.22 | 1.00 | 1.31 | 0.31 | 0.21 |
| FIRE100K-LLaVA | 0.89 | 1.09 | 0.68 | 0.29 | 0.95 | 0.34 | 0.30 | 0.16 | 0.86 | 1.38 | 0.81 | 0.38 | **0.95** | **2.20** | **0.60** | **0.39** |
| FIRE-LLaVA | **0.83** | **1.36** | **0.77** | **0.34** | **0.93** | **0.65** | **0.49** | **0.17** | **0.80** | **1.73** | **1.05** | **0.41** | 0.96 | 2.04 | 0.57 | 0.36 |

| Model | MM-Vet | | | | SEED-Bench | | | | ScienceQA | | | | LLaVA-wild | | | |
|---|---|---|---|---|---|---|---|---|---|---|---|---|---|---|---|---|
| | AT | ADR | ATR | RR | AT | ADR | ATR | RR | AT | ADR | ATR | RR | AT | ADR | ATR | RR |
| LLaVA-NeXT | 1.00 | 0.80 | 0.31 | 0.13 | 1.00 | 2.30 | 0.56 | 0.48 | 1.00 | 2.81 | 0.70 | 0.56 | 1.00 | 0.45 | 0.19 | 0.03 |
| FIRE100K-LLaVA | 0.97 | 0.99 | 0.48 | 0.23 | 0.83 | 3.18 | 0.75 | 0.68 | 0.98 | 2.95 | 0.78 | 0.62 | 0.99 | 0.79 | 0.33 | **0.12** |
| FIRE-LLaVA | **0.87** | **1.18** | **0.60** | **0.26** | **0.81** | **3.34** | **0.84** | **0.69** | **0.83** | **3.94** | **1.08** | **0.78** | 0.96 | 0.85 | 0.50 | 0.12 |

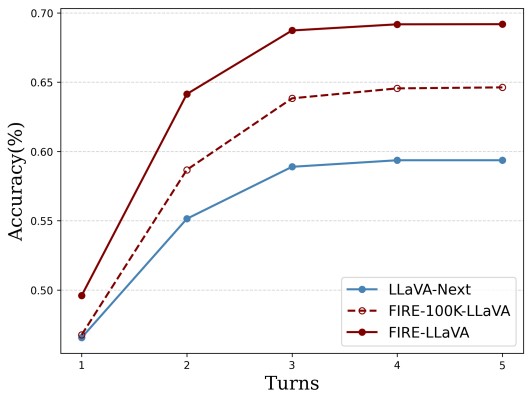

(a) AT(↓)    (b) ADR(↑)

(c) ATR(↑)    (d) RR(↑)

Figure 6: Impact of training set size on model performance.

## 5.5 Ablation Studies

In Fig. 6, we evaluate the feedback-refining ability of VLMs using different amounts of training data from the FIRE dataset. Concretely, we first use the FIRE-100K data. Then, we gradually sample data from FIRE-1M, varying from 200K to 1000K, combined with FIRE-100K to train the LLaVA-NEXT-8B model. Overall, the results indicate that more training data leads to better performance across all evaluated metrics. The substantial improvements, particularly with the initial 100K dialogues and the noted enhancement at around 700K dialogues, demonstrate the high quality of the FIRE dataset and the model's emergent capabilities with more training data.

Figure 7: Accuracy improvement with more dialogue turns.

In Fig. 7, we present the performance curve in FIRE-Bench concerning the number of turns in dialogues, evaluating LLaVA-NeXT, FIRE100K-LLaVA, and FIRE-LLaVA. We report the percentage of correctly answered samples (those scores greater than 8) after each turn. As the number of turns increases, the percentage of correctly answered samples rises across all three models. 46.57% and 46.77% of the test data is correctly answered in the first turn, for the LLaVA-NeXT model and FIRE100K-LLaVA respectively. For FIRE-LLaVA, 49.60% of the data is correctly answered in the first turn, and this increases to 69.19% after five turns, with 19.59% of the samples being corrected based on feedback. Compared to the LLaVA-NeXT model, FIRE-LLaVA shows an additional 6.79% improvement (from 49.60% → 69.19% *vs.* 46.57% → 59.37%), highlighting the effectiveness of FIRE-LLaVA when trained on FIRE.

## 5.6 Visualization

In Fig. 8, we visualize three cases in free dialogues using FIRE-LLaVA and FD-LLaVA. We observe that FIRE-LLaVA can understand the intent behind the feedback and refines its responses according to the feedback on chart understanding, visual concept perception, visual relationship reasoning, and OCR, making efficient user-agent interactions.

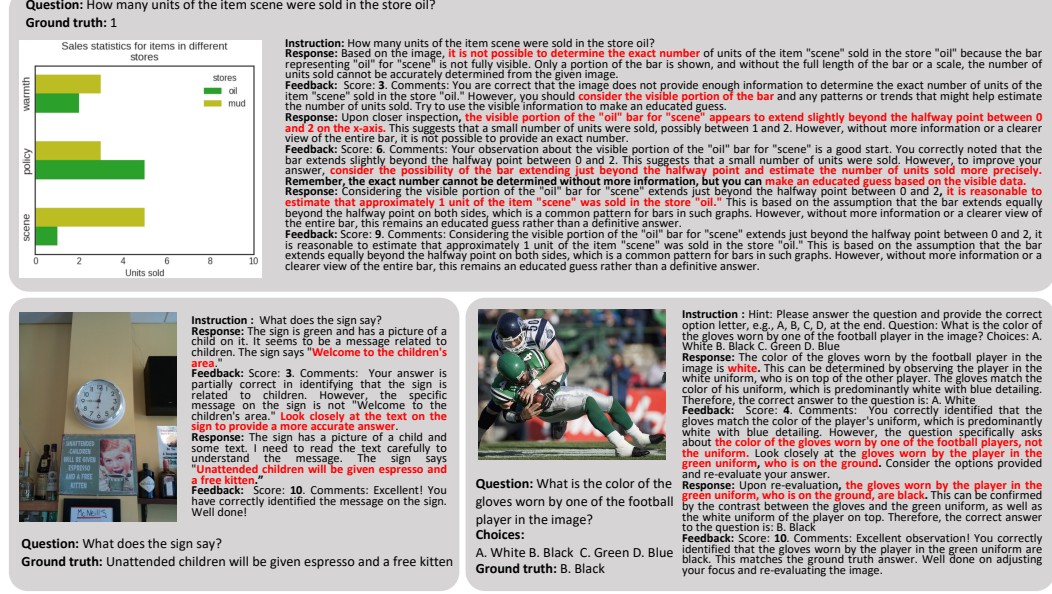

Figure 8: Case study of the feedback-refining ability in our model.

## 6 Conclusion

In this paper, we have presented FIRE, a feedback-refinement dataset with 1.1M multi-turn conversations, which empowers VLMs to refine their responses based on given feedback. Given proper prompts, GPT-4V can produce high-quantity conversations with feedback and responses. Using the 100K GPT-4V generated data as seeds, a student model and a teacher model can freely expand the feedback-refinement data to 1.1M with a similar data quality to GPT-4V. Experiments show that VLMs trained on FIRE have significant improvements in their feedback-refining ability.

**Limitation.** In the current FIRE dataset, the feedback data is limited in the textual form. Practical feedback usually involves diverse multimodal information, such as pointing out image regions. We will further expand FIRE with multimodal feedback data. In addition, although we use a filter process to remove low-quality data, we still cannot completely guarantee the quality of the data. In the future, we will combine human verification with machine verification to improve the quality.

**Acknowledgements.** This work was partly supported by the National Science and Technology Major Project (2022ZD0114900). This work was partly supported by the Natural Science Foundation of China (NSFC) under Grants No. 62176021 and No. 62172041, the Natural Science Foundation of Shenzhen under Grant No. JCYJ20230807142703006, and the Key Research Platforms and Projects of the Guangdong Provincial Department of Education under Grant No.2023ZDZX1034. Mehrtash Harandi is supported by funding from the Australian Research Council Discovery Program DP230101176.

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

# A Human Verification on FIRE

## A.1 Human verification on data quality

To evaluate the data quality of generated data in FIRE-100K, FIRE-1M, and FIRE-Bench, we conduct a user study for the three splits of FIRE. Concretely, we randomly sample 100 conversations from each of the three splits, and ask 10 persons to provide scores (1-5) for feedback and refined responses in each turn of conversations. For the feedback, we ask the person "Please consider the quality of the refined feedback, based on its correctness, relevance, clarity, and constructiveness. Give a score (1-5). 1 means its quality is bad, and 5 means its quality is very good". For the refined response, we ask the person "Please consider the quality of the response, based on its improvement, correctness, and completeness. Given a score (1-5). 1 means its quality is bad, and 5 means its quality is very good". The interface of the user study is shown in Fig. A1. We report the average scores in Tab. A1. We can find that, most users provide high scores for generated data in the three splits, showing that our dataset has high-quality data.

Table A1: Average scores from humans on FIRE-100K, FIRE-1M, and FIRE-Bench, with 5 being the highest score.

| FIRE-100K | | FIRE-1M | | FIRE-Bench | |
|---|---|---|---|---|---|
| Feedback | Response | Feedback | Response | Feedback | Response |
| 4.87 | 4.66 | 4.84 | 4.73 | 4.88 | 4.74 |

## A.2 Human verification on FIRE-LLaVA

To evaluate the models qualitatively, we conducted a human study comparing responses from LLaVA-Next-8B and FIRE-LLaVA. The interface is shown in Fig. A2. We randomly sampled 100 instances and provided each model with identical initial responses and feedback, asking them to generate refined responses. Three independent human evaluators assessed these responses, without knowing which model generated which response (responses were shuffled to ensure blinding). The evaluation results, detailed in Tab. A2, show that FIRE-LLaVA outperforms LLaVA-Next-8B with a significantly higher preference score (37.67 vs. 24.33), indicating that FIRE-LLaVA's responses are more aligned with human preferences.

Table A2: Human evaluation for LLaVA-Next-8B and FIRE-LLaVA.

| | FIRE-LLaVA is Better | Tie | LLaVA-Next-8B is Better |
|---|---|---|---|
| **Votes** | 37.67 | 38 | 24.33 |

# B Data source

Our dataset uses images from 27 diverse sources to provide a robust training dataset for FIRE. All 27 datasets are public datasets, and all the images can be downloaded via links in Tab. A3. The comprehensive list of the source datasets and links to their metadata are detailed below:

Table A3: Data utilized from 27 source datasets for training and test data in FIRE.

| | | | |
|---|---|---|---|
| LLaVA (train) [45] | COCO (train) [41] | SAM (train) [27] | VG (train) [28] |
| Web-Landmark (train) [8] | WikiArt (train) [59] | OCRVQA (train) [54] | AI2D (train) [25] |
| ALLaVA-Vflan (train) [4] | Web-Celebrity (train) [8] | Share-TextVQA (train) [8] | |
| ChartQA (train&test) [51] | DocVQA (train&test) [52] | DVQA (train&test) [24] | GeoQA+ (train&test)[5] |
| VQAV2 (train&test) [18] | GQA (train&test) [22] | TextVQA (train&test) [61] | Synthdog-EN (train&test)[26] |
| LLaVA-in-the-Wild (test) [45] | MMMU (test) [68] | MME (test) [16] | MM-Vet (test) [67] |
| MathVerse (test) [70] | MathVista (test) [47] | ScienceQA (test) [49] | SEED-bench (test) [32] |

Please evaluate the quality of the student model's response and the teacher model's feedback. Rate from 1 to 5, where a higher score indicates better quality. You can refer to the following criteria when scoring:

**Evaluation Criteria for Student Model's Response**

1. **Improvement:** Evaluate the improvement based on the teacher's feedback in the student's answer.
2. **Correctness:** Assess the accuracy of the student's answer and its alignment with known facts or ground truth.
3. **Relevance:** Evaluate if the student's response directly addresses the question and if it is free of omissions or off-topic content.
4. **Completeness:** Assess if the student's response is comprehensive, including necessary details and information.

**Evaluation Criteria for Teacher Model's Feedback**

1. **Correctness:** Evaluate if the feedback is based on actual errors or deficiencies in the student's answer and if it aligns with known ground truth.
2. **Relevance:** Assess if the feedback directly addresses the student's answer without introducing irrelevant information and if it relates to the specific question the student answered.
3. **Clarity:** Determine if the feedback is clear and explicit enough for the student model to understand and improve upon, using concise and easy-to-understand language.
4. **Constructiveness:** Evaluate if the feedback provides specific advice that will help the student improve and if it points out how to avoid similar mistakes or improve the answer.

## Case 2

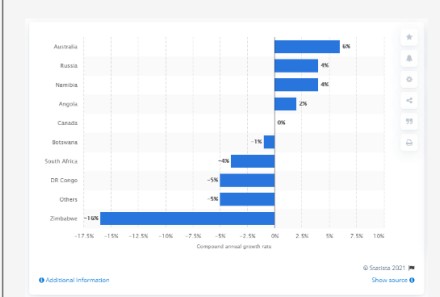

**Question:** What is the difference between Australia and Angola Compound annual growth rate for the top diamond producing country world wide between 2010 and 2015?

**Groundtruth:** 4

**Student:** Thought: I see various countries listed with their compound annual growth rates. I need to find the rates for Australia and Angola and calculate the difference. I think Australia's rate is higher than Angola's. Answer: The difference is 2%.

**Teacher:** Score: 4. Feedback: You correctly identified the need to find the compound annual growth rates for both Australia and Angola and calculate the difference. However, your calculation seems to be incorrect. Please check the rates for both countries again and ensure you are calculating the difference accurately. Remember to subtract the smaller rate from the larger rate to find the correct difference.

○ 1 - Very Poor  ○ 2 - Poor  ○ 3 - Average  ○ 4 - Good  ○ 5 - Excellent

**Student:** Thought: I need to recheck the values for both countries. I see that Australia has a growth rate of 6% and Angola has a rate of 2%. To find the difference, I should subtract Angola's rate from Australia's rate. Answer: The difference is 4%.

○ 1 - Very Poor  ○ 2 - Poor  ○ 3 - Average  ○ 4 - Good  ○ 5 - Excellent

**Teacher:** Score: 10. Feedback: Excellent work! You have correctly identified the compound annual growth rates for both Australia and Angola and accurately calculated the difference between them. Your answer is now correct and well-presented. Great job on adjusting your approach and achieving the correct result.

○ 1 - Very Poor  ○ 2 - Poor  ○ 3 - Average  ○ 4 - Good  ○ 5 - Excellent

Save Scores and Next Case

Figure A1: The screenshot of the interface for the human verification on data quality.

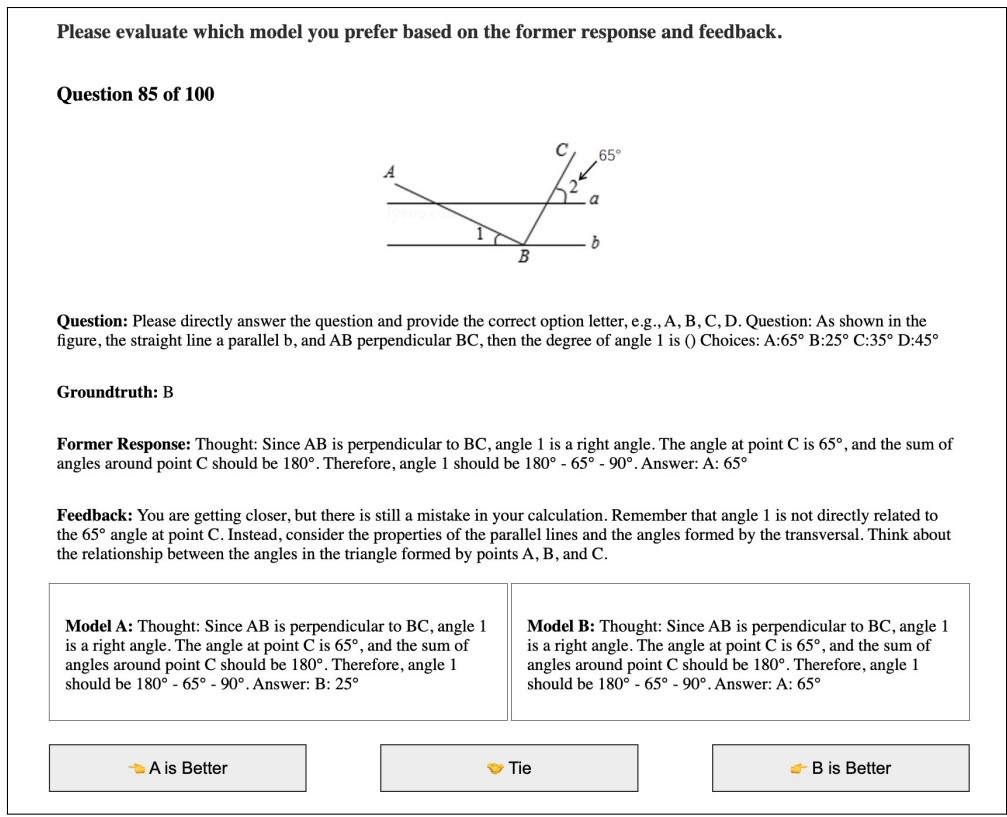

Figure A2: The interface of human evaluation on FIRE LLaVA. We randomly sampled 100 instances, allowing both the baseline and FIRE-LLaVA models to generate inferences under identical former responses and feedback conditions. Three participants were invited to rate the responses. For each sample, the responses were shuffled to randomize the association between the responses and Model A/B. Participants selected their preference by choosing 'A is better,' 'Tie,' or 'B is better' based on their judgment.

## C    Additional Experimental Results

### C.1    Error bar

We report the error bar of average turn (AT), average dialogue refinement (ADR), average turn refinement (ATR), and refinement ratio (RR) in fixed dialogues. We run the model three times and compute the standard deviation, as shown in Tab. A4. Comparisons among the four metrics, the standard deviation is relatively small, less than $8\%$ of the average results, showing that our method can achieve stable feedback-refining ability.

Table A4: Results in free dialogue over all test data in FIRE.

| Model | AT ($\downarrow$) | ADR ($\uparrow$) | ATR ($\uparrow$) | RR ($\uparrow$) |
|---|---|---|---|---|
| LLaVA-Next-8B | 1 | 0.97 | 0.41 | 0.25 |
| FIRE100K-LLaVA-8B | $0.92 \pm 0.026$ | $1.27 \pm 0.013$ | $0.55 \pm 0.042$ | $0.34 \pm 0.022$ |
| FIRE-LLaVA-8B | $\mathbf{0.84 \pm 0.015}$ | $\mathbf{1.56 \pm 0.012}$ | $\mathbf{0.66 \pm 0.053}$ | $\mathbf{0.39 \pm 0.028}$ |

### C.2    More VLMs

#### C.2.1    LLaVA-Next-Vicuna-7B

We further train a FIRE-LLaVA-Vicuna model that replaces LLaMA3-8B in FIRE-LLaVA with Vicuna1.5-7B. Results are shown in Tab. A5. Results of using Vicuna1.5-7B demonstrate the effectiveness of FIRE again, where FIRE-LLaVA-Vicuna has better feedback-refining ability than the

Table A5: Results of FIRE-LLaVA-Vicuna in free dialogue over all test data in FIRE.

| Model | AT ($\downarrow$) | ADR ($\uparrow$) | ATR ($\uparrow$) | RR ($\uparrow$) |
|---|---|---|---|---|
| LLaVA-Next-Vicuna | 1.00 | 0.98 | 0.49 | 0.24 |
| FIRE-LLaVA-Vicuna | **0.94** | **1.11** | **0.57** | **0.27** |

Table A6: Results of LLaVA-1.5-Vicuna-7B and LLaVA-1.5-Vicuna-7B-FIRE on FIRE-Bench.

| Model | AT ($\downarrow$) | ADR ($\uparrow$) | ATR ($\uparrow$) | RR ($\uparrow$) |
|---|---|---|---|---|
| LLaVA-1.5-Vicuna | 1.00 | 0.62 | 0.46 | 0.12 |
| FIRE-LLaVA-1.5 | **0.94** | **0.80** | **0.61** | **0.20** |

original LLaVA-Next-Vicuna model on AT, ADR, ATR, and RR, showing the helpfulness for the feedback-refining ability.

### C.2.2 LLaVA-1.5-Vicuna-7B

We have also performed experiments using LLaVA-1.5-Vicuna-7B as another baseline. The results are presented in Tab. A6. The findings demonstrate that the LLaVA-1.5-Vicuna-7B model fine-tuned on FIRE100K outperforms the original LLaVA-1.5-Vicuna-7B model across all four metrics—AT, ADR, ATR, and RR—highlighting the effectiveness of our FIRE dataset.

### C.3 More Visualization

We provide more cases tested on the FIRE-Bench to show the feedback-refining ability, as shown in Figs. A3 to A10. We observe that FIRE-LLaVA trained on FIRE has good feedback-refining ability on different tasks, such as visual perception, visual question answering, image captioning, OCR reasoning, document understanding, math reasoning, science question answering, and chart analysis.

## D Prompt

### D.1 Prompt for GPT-4V

We show examples of prompts for GPT-4V to generate feedback-refinement conversations, as shown in Fig. A11 and Fig. A12.

### D.2 Prompt for Student and Teacher models

We show examples of prompts for student and teacher models to simulate feedback-refinement conversations, as shown in Fig. A13 and Fig. A14, respectively. In Fig. A13, the prompt for the student model to generate $n$-th response is shown. The prompt contains the last $n-1$ rounds' student responses and the textual comments from the teacher model. The prompt for the teacher model is shown in Fig. A14. Firstly, the prompt provides user instruction and ground truth. Secondly, the prompt contains instructions that format the teacher model's feedback as textual comments and numeric scores. Finally, the prompt incorporates the only latest student response into its context.

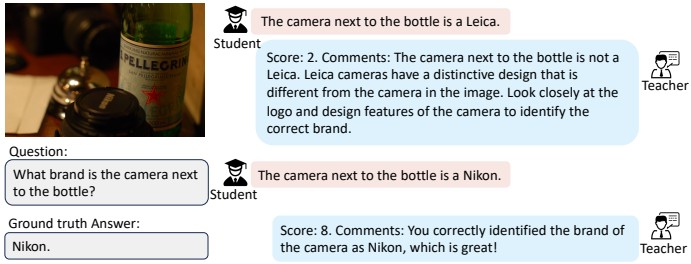

Figure A3: One case for feedback-refining ability in OCR.

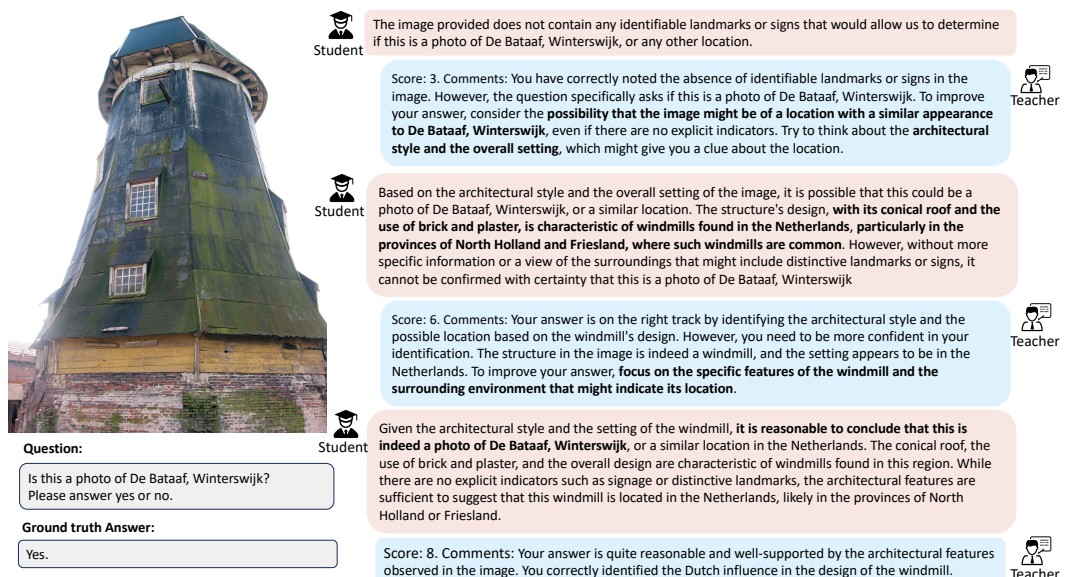

Figure A4: One case for feedback-refining ability in visual perception.

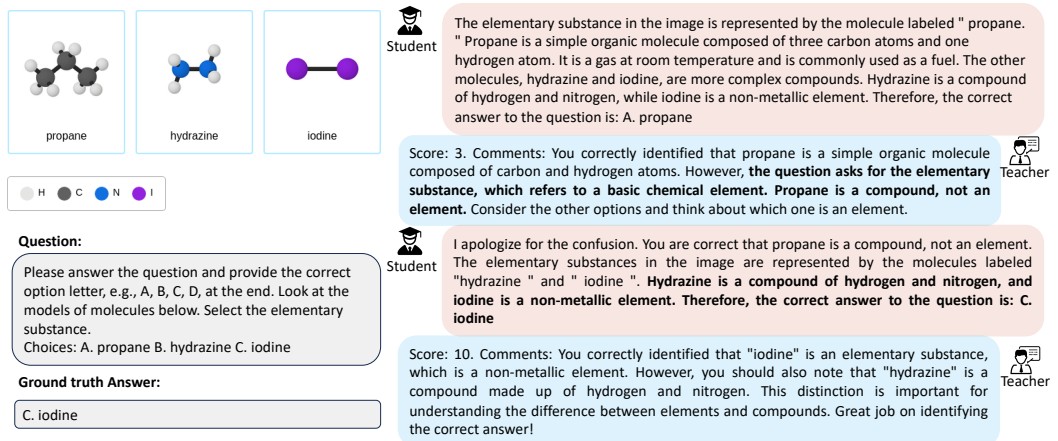

Figure A5: One case for feedback-refining ability in science question answering.

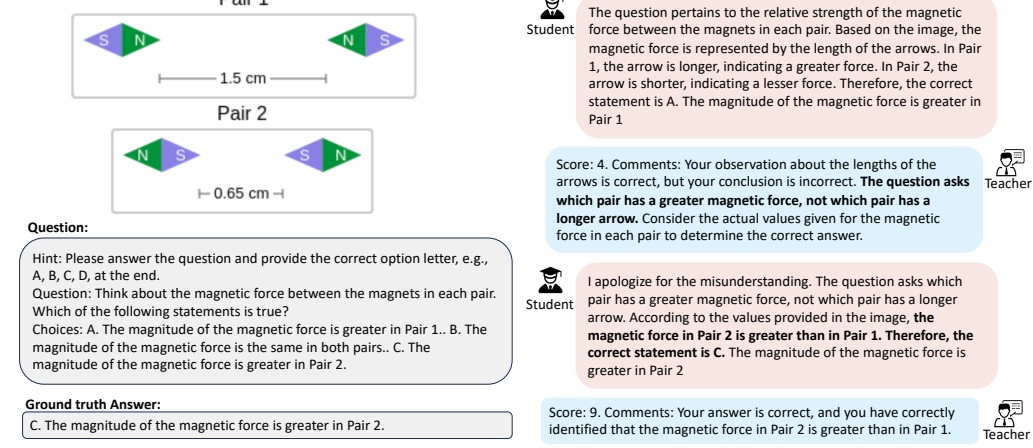

Figure A6: One case for feedback-refining ability in science question answering.

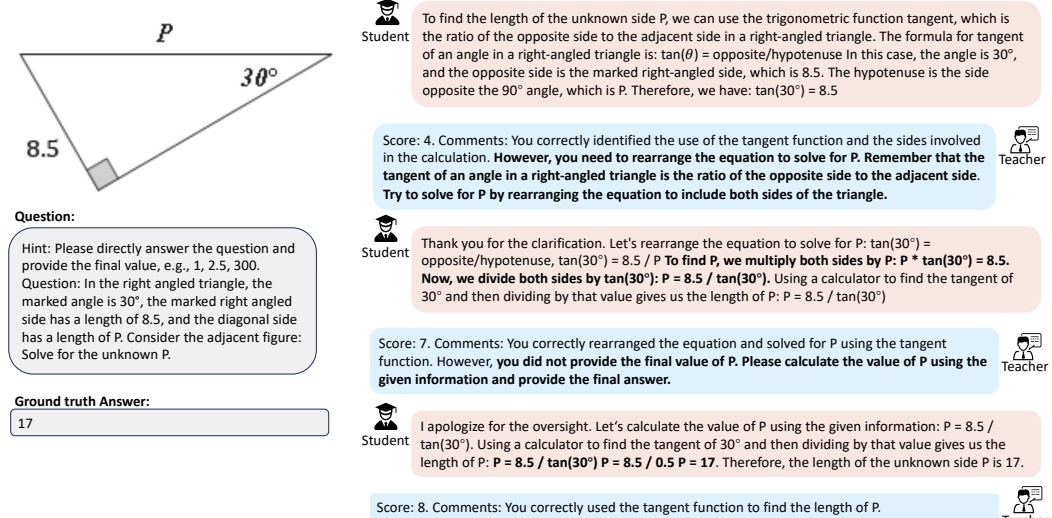

Figure A7: One case for feedback-refining ability in math reasoning.

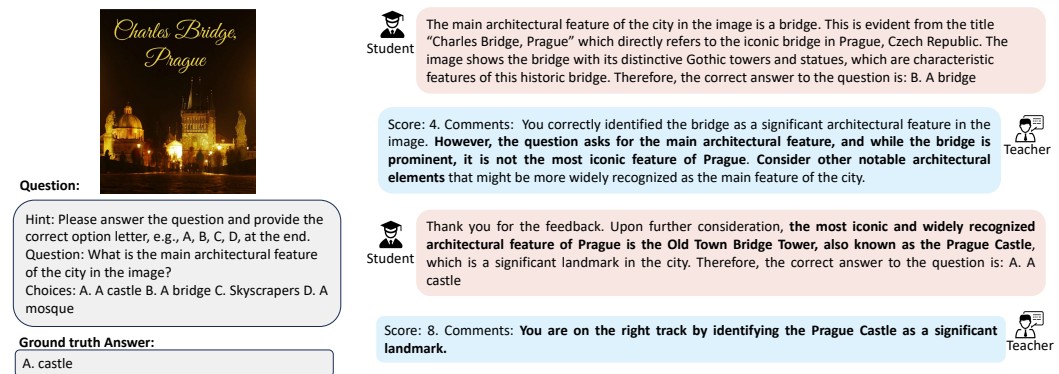

Figure A8: One case for feedback-refining ability in visual question answering.

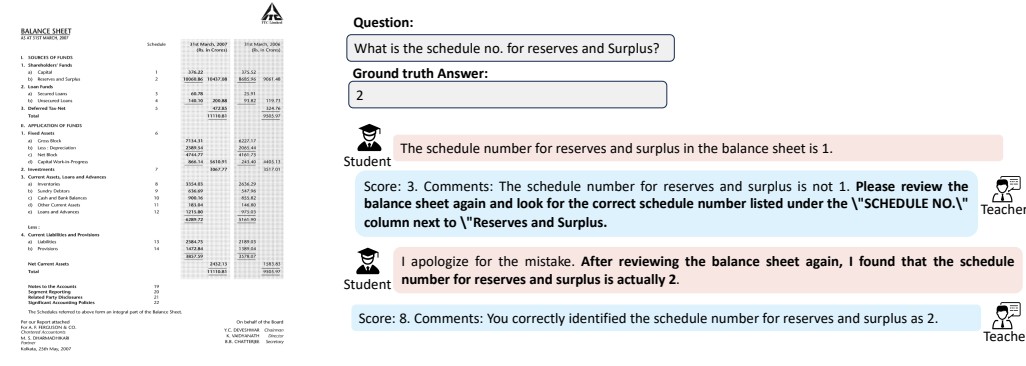

Figure A9: One case for feedback-refining ability in document understanding.

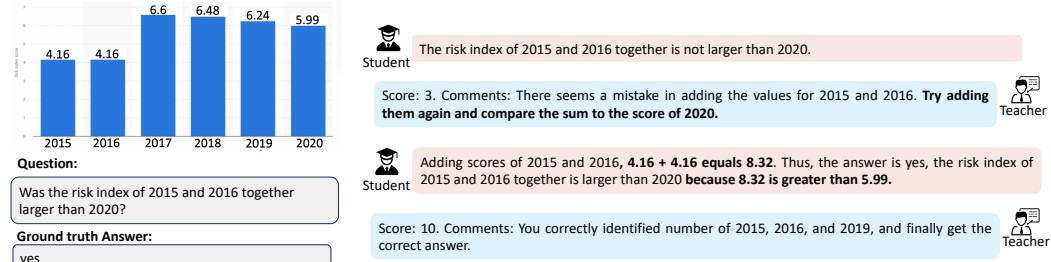

Figure A10: One case for feedback-refining ability in chart analysis.

You are a helpful assistant that can generate a dialogue between a teacher and a student. The student is trying to answer a question about an image. The student first gives a wrong answer. Based on the given groundtruth answer, the teacher provides feedback to help the student gradually improve its answer. Use the following template to generate the dialogue:

"""
# Round 1

## Student's response
Thought: <here is the student's thought process about the question. Do NOT use the words "teacher" or "student". >
Answer: <here is the student's answer to the question.>

## Teacher's feedback
Score: <compare the student's answer with the groundtruth answer in terms of accuracy, relevance, helpfulness, and level of detail. Provide an overall score on a scale of 1 to 10, where a higher score indicates better overall performance.>
Feedback: <provide feedback on the student's answer. Do not directly tell the groundtruth answer. The feedback should identify which parts of the student's answer are incorrect, what is missing in the student's answer, and how to improve the student's answer.>

# Round 2
...

# Round n
...

"""
The number of rounds should depend on the difficulty of the question. More rounds should be used for difficult questions, while fewer rounds should be used for easy questions.

Figure A11: System prompt for GPT-4V for Student-Teacher conversation generation.

Here are the given image, question: <question> and groundtruth answer: <groundtruth>, now generate a dialogue:

Figure A12: User prompt for GPT-4V for Student-Teacher conversation generation.

You are a helpful language and vision assistant. You are able to understand the visual content that the user provides, and assist the user with a variety of tasks using natural language
<user_instruction>

**# Round 1**
<student_response_round_1>
<feedback_round_1>

...
**# Round n-1**
<student_response_round_n-1>
<feedback_round_n-1>

Based on the feedback, answer the question again:

Figure A13: Prompt for student model to simulate feedback-refinement conversations.

You are a helpful language and vision assistant. You are able to understand the visual content that the user provides, and assist the user with a variety of tasks using natural language

Question: <question>
Groundtruth: <groundtruth>

Please compare my answer with the groundtruth answer and provide helpful, detailed, and polite feedback to help me improve my answer. Formulate the feedback as:
"""
Score: <compare the provided response with the groundtruth answer in terms of accuracy, relevance, helpfulness, and level of detail, and provide an overall score on a scale of 1 to 10, where a higher score indicates better overall performance.>

Feedback: <provide feedback on the response. Do NOT directly tell the groundtruth answer. The feedback should identify which parts of my answer are incorrect, what is missing in the response, and how to improve the response.>
"""
Here is the student response: <student_response>, now please provide the feedback:

Figure A14: Prompt for teacher model to simulate feedback-refinement conversations.

# E   Potential Negative Societal Impacts

The VLMs may generate harmful outputs based on human induction feedback, resulting in risks, such as false information, discrimination, violent and pornographic content, and privacy leaks *etc*. To mitigate the risks of these harmful outputs, we will strictly filter and review the model outputs based on feedback in the future. In addition, users may become overly dependent on the model's outputs given feedback, neglecting the need for independent thinking and verification of information.

