# A    Potential Negative Societal Impacts

The VLMs may generate harmful outputs based on human induction feedback, resulting in risks, such as false information, discrimination, violent and pornographic content, and privacy leaks *etc*. To mitigate the risks of these harmful outputs, we will strictly filter and review the model outputs based on feedback in the future. In addition, users may become overly dependent on the model's outputs given feedback, neglecting the need for independent thinking and verification of information.

# B    Human Verification on FIRE

To evaluate the data quality of generated data in FIRE-100K, FIRE-1M, and FIRE-Bench, we conduct a user study for the three splits of FIRE. Concretely, we randomly sample 100 conversations from each of the three splits, and ask 10 persons to provide scores (1-5) for feedback and refined responses in each turn of conversations. For the feedback, we ask the person "Please consider the quality of the refined feedback, based on its correctness, relevance, clarity, and constructiveness. Give a score (1-5). 1 means its quality is bad, and 5 means its quality is very good". For the refined response, we ask the person "Please consider the quality of the response, based on its improvement, correctness, and completeness. Given a score (1-5). 1 means its quality is bad, and 5 means its quality is very good". The interface of the user study is shown in  Fig. A1. We report the average scores in Tab. A1. We can find that, most users provide high scores for generated data in the three splits, showing that our dataset has high-quality data.

Table A1: Average scores from humans on FIRE-100K, FIRE-1M, and FIRE-Bench, with 5 being the highest score.

| FIRE-100K | | FIRE-1M | | FIRE-Bench | |
|---|---|---|---|---|---|
| Feedback | Response | Feedback | Response | Feedback | Response |
| 4.87 | 4.66 | 4.84 | 4.73 | 4.88 | 4.74 |

# C    Additional Experimental Results

## C.1    Error bar

We report the error bar of average turn (AT), average dialogue refinement (ADR), average turn refinement (ATR), and refinement ratio (RR) in fixed dialogues. We run the model three times and compute the standard deviation, as shown in Tab. A2. Comparisons among the four metrics, the standard deviation is relatively small, less than $8\%$ of the average results, showing that our method can achieve stable feedback-refining ability.

Table A2: Results in free dialogue over all test data in FIRE.

| Model | AT ($\downarrow$) | ADR ($\uparrow$) | ATR ($\uparrow$) | RR ($\uparrow$) |
|---|---|---|---|---|
| LLaVA-Next-8B | 1 | 0.97 | 0.41 | 0.25 |
| FIRE100K-LLaVA-8B | $0.92 \pm 0.026$ | $1.27 \pm 0.013$ | $0.55 \pm 0.042$ | $0.34 \pm 0.022$ |
| FIRE-LLaVA-8B | $\mathbf{0.84 \pm 0.015}$ | $\mathbf{1.56 \pm 0.012}$ | $\mathbf{0.66 \pm 0.053}$ | $\mathbf{0.39 \pm 0.028}$ |

## C.2    Performance Curves with respect to data number

We have a total of 1.1M training data in FIRE. We evaluate the performance of VLMs using different number data in FIRE. In Fig. A2, we present the curves of AT, ATR, ATR, and RR using different numbers of training data in FIRE. Concretely, we first use the FIRE-100K data. Then, we randomly sample data from FIRE-100K, varying from 100K to 1000K, combined with FIRE-100K to train the LLaVA-NEXT-8B model. Results show that more data leads to better performance. FIRE-100K data brings a significant improvement, and the performance continues to grow with data increases. Then, the performance slightly increases, and finally achieves obviously better feedback-refining ability than the original LLaVA-Next-8B model. This experiment shows the quality of data in FIRE again.

Please evaluate the quality of the student model's response and the teacher model's feedback. Rate from 1 to 5, where a higher score indicates better quality. You can refer to the following criteria when scoring:

**Evaluation Criteria for Student Model's Response**

1. **Improvement:** Evaluate the improvement based on the teacher's feedback in the student's answer.
2. **Correctness:** Assess the accuracy of the student's answer and its alignment with known facts or ground truth.
3. **Relevance:** Evaluate if the student's response directly addresses the question and if it is free of omissions or off-topic content.
4. **Completeness:** Assess if the student's response is comprehensive, including necessary details and information.

**Evaluation Criteria for Teacher Model's Feedback**

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

Figure A13: User prompt for GPT-4V for Student-Teacher conversation generation.

prompt contains instructions that format the teacher model's feedback as textual comments and numeric scores. Finally, the prompt incorporates the only latest student response into its context.

# E Datasheet for FIRE

We present a Datasheet [16] for documentation and responsible usage of our internet knowledge databases. The required author statement, hosting, licensing, metadata, and maintenance plan can be found in the datasheet.

## E.1 Motivation

**For what purpose was the dataset created?** We create this large-scale dataset to facilitate research towards enhancing the feedback-reflecting ability of VLMs.

**Who created the dataset (e.g., which team, research group) and on behalf of which entity (e.g., company, institution, organization)?** This dataset was created by Pengxiang Li (Beijing Institute

You are a helpful language and vision assistant. You are able to understand the visual content that the user provides, and assist the user with a variety of tasks using natural language
<user_instruction>

# Round 1
<student_response_round_1>
<feedback_round_1>

...
# Round n-1
<student_response_round_n-1>
<feedback_round_n-1>

Based on the feedback, answer the question again:

Figure A14: Prompt for student model to simulate feedback-refinement conversations.

You are a helpful language and vision assistant. You are able to understand the visual content that the user provides, and assist the user with a variety of tasks using natural language

Question: <question>
Groundtruth: <groundtruth>

Please compare my answer with the groundtruth answer and provide helpful, detailed, and polite feedback to help me improve my answer. Formulate the feedback as:
"""
Score: <compare the provided response with the groundtruth answer in terms of accuracy, relevance, helpfulness, and level of detail, and provide an overall score on a scale of 1 to 10, where a higher score indicates better overall performance.>

Feedback: <provide feedback on the response. Do NOT directly tell the groundtruth answer. The feedback should identify which parts of my answer are incorrect, what is missing in the response, and how to improve the response.>
"""
Here is the student response: <student_response>, now please provide the feedback:

Figure A15: Prompt for teacher model to simulate feedback-refinement conversations.

of Technology), Zhi Gao (BIGAI), Bofei Zhang (BIGAI), Tao Yuan (BIGAI), Yuwei Wu (Beijing Institute of Technology), Mehrtash Harandi (Monash University), Yunde Jia (Beijing Institute of Technology), Song-chun Zhu (BIGAI), Qing Li (BIGAI).

## E.2 Distribution

**Will the dataset be distributed to third parties outside of the entity (e.g., company, institution, organization) on behalf of which the dataset was created?** Yes, the dataset is publicly available on the internet.

**How will the dataset will be distributed (e.g., tarball on website, API, GitHub)?** The dataset can be downloaded from https://huggingface.co/datasets/PengxiangLi/FIRE. We use Creative Commons Attribution 4.0 License (CC BY 4.0). The Croissant metadata can be found on the dataset hosting platform (https://huggingface.co/).

**Have any third parties imposed IP-based or other restrictions on the data associated with the instances?** No.

**Do any export controls or other regulatory restrictions apply to the dataset or to individual instances?** No.

### E.3 Maintenance

**Who will be supporting/hosting/maintaining the dataset?** The authors will be supporting, hosting, and maintaining the dataset.

**How can the owner/curator/manager of the dataset be contacted (e.g., email address)?** Please contact Qing Li (liqing@bigai.ai).

**Is there an erratum?** No. We will make announcements if there is any.

**Will the dataset be updated (e.g., to correct labeling errors, add new instances, delete instances)?** Yes. New updates will be posted on https://mm-fire.github.io/.

**If the dataset relates to people, are there applicable limits on the retention of the data associated with the instances (e.g., were the individuals in question told that their data would be retained for a fixed period of time and then deleted)?** The images in our dataset might contain human subjects, but all of them are from public datasets.

**Will older versions of the dataset continue to be supported/hosted/maintained?** Yes, old versions will be permanently accessible on huggingface.co.

**If others want to extend/augment/build on/contribute to the dataset, is there a mechanism for them to do so?** Yes, please refer to https://mm-fire.github.io/.

### E.4 Composition

**What do the instances that comprise the dataset represent?** Our data is generally stored in the json files. Every instance includes the path of an image and the feedback-reflecting dialogues.

**How many instances are there in total (of each type, if appropriate)?** There are 1.2M samples (1.1M for training set, 0.1M for test set), among which 200K are GPT-4V generated data, while the rests are simulated via the FIRE-LLaVA and FIRE-LLaVA-FD.

**Does the dataset contain all possible instances or is it a sample (not necessarily random) of instances from a larger set?** We provide all instances in our Huggingface data repositories.

**Is there a label or target associated with each instance?** Yes.

**Is any information missing from individual instances?** No.

**Are relationships between individual instances made explicit (e.g., users' movie ratings, social network links)?** No.

**Are there recommended data splits (e.g., training, development/validation, testing)?** Yes. FIRE-100K and FIRE-1M are used for training and FIRE-Bench is used for testing.

**Are there any errors, sources of noise, or redundancies in the dataset?** Please refer to the limitations in Sec. 6.

**Is the dataset self-contained, or does it link to or otherwise rely on external resources (e.g., websites, tweets, other datasets)?** The dataset is self-contained.

**Does the dataset contain data that might be considered confidential?** No.

**Does the dataset contain data that, if viewed directly, might be offensive, insulting, threatening, or might otherwise cause anxiety?** No.

### E.5 Collection Process

The collection procedure, preprocessing, and cleaning are explained in detail in Section 2 of the main paper.

**Who was involved in the data collection process (e.g., students, crowdworkers, contractors) and how were they compensated (e.g., how much were crowdworkers paid)?** All data collection, curation, and filtering are done by FIRE coauthors.

**Over what timeframe was the data collected?** The data was collected between Jan. 2024 and May 2024.

### E.6 Uses

**Has the dataset been used for any tasks already?** Yes, we have used FIRE for training our VLMs, including FIRE-LLaVA-Vicuna, FIRE100K-LLaVA, FIRE-LLaVA, FIRE-LLaVA-FD.

**What (other) tasks could the dataset be used for?** Our dataset is primarily for facilitating research in enhancing the feedback-reflecting ability of VLMs. Our data might also be used to benchmark existing and future VLMs.

**Is there anything about the composition of the dataset or the way it was collected and preprocessed/cleaned/labeled that might impact future uses?** No.

**Are there tasks for which the dataset should not be used?** We strongly oppose any research that intentionally generates harmful or toxic content using our data.

## F   Data source

Our dataset uses images from 27 diverse sources to provide a robust training dataset for FIRE. All 27 datasets are public datasets, and all the images can be downloaded via links in Tab. A4. The comprehensive list of the source datasets and links to their metadata are detailed below:

Table A4: Data utilized from 27 source datasets for training and test data in FIRE.

| | | | | |
|---|---|---|---|---|
| LLaVA [41] | COCO [37] | SAM [26] | VQAV2 [17] | GQA [21] |
| VG [27] | Web-Celebrity [8] | Web-Landmark [8] | WikiArt [54] | ALLaVA-Vflan [4] |
| ChartQA [46] | DocVQA [47] | DVQA [23] | GeoQA+[5] | Synthdog-EN[25] |
| LLaVA-in-the-Wild [41] | MMMU [63] | MME [14] | MM-Vet [62] | SEED-bench [30] |
| OCRVQA [49] | TextVQA [56] | Share-TextVQA [8] | AI2D [24] | MathVerse [65] |
| MathVista [43] | ScienceQA [45] | | | |