# OpenReview forum: "FIRE: A Dataset for Feedback Integration and Refinement Evaluation of Multimodal Models"
_NeurIPS.cc/2024/Datasets_and_Benchmarks_Track — NeurIPS 2024 Track Datasets and Benchmarks Poster_

### Official Review · Reviewer_2z6w · 2024-07-27
**Comprehensive framework for feedback integration and refinement evaluation**

**Rating:** 8
**Confidence:** 3
**Correctness:** As far as I can see the claims made a…
**Clarity:** The paper is written well.

**Review:**

The quality of the paper is good. It has sufficient clarity and the idea of feedback and refinement integration is original and significant for a lot of real-world applications (also in other language models).

**Strengths:**

The data set creation is described in detail, and the data is comprised of various tasks and settings. The idea of the evaluation and training process is very convincing.

**Additional Feedback:**

None.

**Documentation:**

Details about data collection/generation are available, and the data has been nicely analyzed for further reference. The data is also available online for further research.

**Ethics:**

I don't suspect any ethical concerns.

**Limitations:**

As the data set is very large, there is only little human curation.

**Opportunities For Improvement:**

The authors mention the possible range for scores only very late in the paper, making it difficult to grasp choices like 10 for the highest difference or 8 for the threshold of a good score.

**Relation To Prior Work:**

The related work section provides enough distinction to the paper at hand.

**Summary And Contributions:**

The paper suggests a framework and a data set to integrate feedback into the training of models and evaluate the refinement gained from the feedback for the model's output. The application realm is VLMs, and the data set is created using generative models.

---

> ### Author Rebuttal · Authors · 2024-08-13
>
> Thank you for your positive feedback on our work, especially the recognition of the originality and significance of our approach. We will address your suggestion and concern below.
>
> > *Opportunities For Improvement: The authors mention the possible range for scores only very late in the paper, making it difficult to grasp choices like 10 for the highest difference or 8 for the threshold of a good score.*
>
> **A:** We will revise the paper to present the range and threshold for the scores earlier (in the Introduction Section), making the rationale behind the scoring choices clearer and easier to understand.
>
> > *Limitation: As the data set is very large, there is only little human curation.*
>
> **A:** Our focus has been on developing an automated data curation scheme that ensures high-quality large-scale datasets. This approach has been widely adopted in previous works, such as LLaVA [a], ShareGPT4V [b] and MiniGPT-4 [c]. To assess data quality, we conducted user studies where 300 randomly selected samples were human-verified, as detailed in Section A of the Appendix. Our dataset consistently achieved near-perfect scores in the user studies, indicating its suitability for training. Therefore, extensive human curation is unnecessary, as it would add significant costs without substantial benefits.
>
> [a] Liu et al. Visual Instruction Tuning. Neurips 2023.
>
> [b] Chen et al. Sharegpt4v: Improving large multi-modal models with better captions. 2023.
>
> [c] Zhu et al. MiniGPT-4: Enhancing Vision-Language Understanding with Advanced Large Language Models. ICLR 2024.

---

### Official Review · Reviewer_q6Ko · 2024-08-15
**A Dataset for Feedback Integration and Refinement Evaluation of Multimodal Models**

**Rating:** 7
**Confidence:** 3

**Review:**

This work is significant for introducing a feedback-refinement dataset that contains a diverse range of tasks. Additionally, the authors propose FIRE-Bench, a benchmark containing 11K feedback-refinement conversations derived from 8 seen and 8 unseen datasets, designed to evaluate the feedback-refining capabilities of VLMs. The paper also demonstrates the effectiveness of fine-tuning the LLaVA model on the FIRE dataset, leading to a marked improvement in feedback-refining capabilities measured by FIRE-Bench.  However, there are a few areas where additional information or clarification could be beneficial.

**Strengths:**

- The paper introduces a feedback-refinement dataset that integrates data from 27 diverse sources, offering a valuable resource for instruction tuning and evaluating VLMs.
- The authors present FIRE-Bench, an evaluation benchmark that spans both seen and unseen datasets, enabling a comprehensive assessment of feedback-refinement capabilities in VLMs.

**Additional Feedback:**

N/A

**Clarity:**

- It appears there are 26 datasets used for the FIRE benchmark: 18 for FIRE-100K/FIRE-1M and 8 for unseen datasets in FIRE-Bench.
- In Figure 1, highlighting the feedback-refinement component of GPT-4V could better illustrate the similarity between the FIRE model and GPT-4V.
- For Figure 2, consistent coloring for each label would enhance readability.
- For Figure 3, the instruction in the gray part should be reframed as a question for clarity.

**Correctness:**

This manuscript does not appear to contain any clear errors in data construction.

**Documentation:**

The data collection pipeline is clear.

**Limitations:**

There are no significant limitations to highlight beyond what the authors have mentioned.

**Opportunities For Improvement:**

- In Table 1, the performance drop in 5 benchmarks is concerning, as 3 out of 5 show significant declines. It would be helpful to discuss why these drops occur.
- Why did you choose to use only one baseline model (LLaVA-Next-8B)? Including more baselines could strengthen the evaluation.
- Why did you use different criteria for filtering dialogues in FIRE-100K and FIRE-1M? For example:
  - FIRE-100K: Filtering dialogues with no score improvements or more than 6 turns.
  - FIRE-1M: Filtering based on feedback ≥ 8 or the number of turns exceeding 3.
- T_baseline Value: What is the value of T_baseline? Since the maximum number of turns is set as n_max = 5, AT should be lower than 1 if T_baseline ≥ 5.
- In the paper, you mention that if n_k = 1, it is removed from the K samples to compute AT, ADR, ATR, and RR. This implies that K might differ across models. Providing the proportion of n_k = 1 per model would be helpful.

**Relation To Prior Work:**

The relation to prior research is also well described.

**Summary And Contributions:**

The paper introduces FIRE, a feedback-refinement dataset consisting of 1.1M multi-turn conversations. FIRE comprises two subsets: FIRE-100K and FIRE-1M. FIRE-100K is generated by GPT-4V, while FIRE-1M is freely generated by models trained on FIRE-100K. The authors also develop FIRE-Bench to evaluate the feedback-refining capabilities of Vision-Language Models (VLMs). By fine-tuning the LLaVA model on their dataset, they demonstrate a significant improvement in feedback-refining capabilities on FIRE-Bench, outperforming the performance observed before fine-tuning.

---

> ### Author Rebuttal · Authors · 2024-08-20
>
> # Rebuttal [1/2]
> Thank you for your time and your insightful and thorough review. We will address your suggestions and concerns below.
>
> ### Opportunities For Improvement:
> > *Q1: In Table 1, the performance drop in 5 benchmarks is concerning, as 3 out of 5 show significant declines. It would be helpful to discuss why these drops occur.*
>
>
> **A:** The performance drop is primarily due to the fact that FIRE-LLaVA did not utilize the "any resolution" technique [a] during training, whereas LLaVA-Next-8B did. The "any resolution" technique involves cropping images into various resolutions and feeding these different image segments into the model to enhance perception and reasoning capabilities. Consequently, FIRE-LLaVA underperformed on benchmarks that require detailed analysis of local regions, such as TextVQA.
>
> To address this, we have now incorporated the "any resolution" technique, identical to the approach used by LLaVA-Next-8B. The updated results, shown in Table A, demonstrate that FIRE-LLaVA’s performance is now comparable to LLaVA-Next-8B. We will include these updated findings in the revised version of the paper.
>
>
> **Table A: Updated results (%) of FIRE-LLaVA and LLaVA-Next-8B on several benchmarks.**
> |	|TextVQA|	MMVet	|LLAVA(W)|
> |-|-|-|-|
> | FIRE-LLaVA (w/o any res) | 61.0 | 38.3 | 73.4|
> | FIRE-LLaVA (w/ any res) | 68.7 | 44.8 | 75.5|
> | LLaVA-Next-8B (w/ any res) | 69.8 | 44.9 | 78.5 |
>
>
> [a] Li et al. LLaVA-NeXT: What Else Influences Visual Instruction Tuning Beyond Data? https://llava-vl.github.io/blog/2024-05-25-llava-next-ablations
>
>
> > *Q2: Why did you choose to use only one baseline model (LLaVA-Next-8B)? Including more baselines could strengthen the evaluation.*
>
>
> **A:** We have conducted experiments with an additional baseline model, LLaVA-NeXT-Vicuna-7B. Due to space constraints, the results and analysis of LLaVA-NeXT-Vicuna-7B were included in the Supplementary Material (refer to 'Section C.4 More VLMs' for details). In the revised version, we plan to integrate this subsection from the appendix into the main paper to provide a more comprehensive evaluation.
>
> Furthermore, we have also performed experiments using LLaVA-1.5-Vicuna-7B as another baseline. The results are presented in Table B. The findings demonstrate that the LLaVA-1.5-Vicuna-7B model fine-tuned on FIRE100K outperforms the original LLaVA-1.5-Vicuna-7B model across all four metrics—AT, ADR, ATR, and RR—highlighting the effectiveness of our FIRE dataset.
>
> **Table B: Results of LLaVA-1.5-Vicuna-7B and LLaVA-1.5-Vicuna-7B-FIRE on FIRE-Bench.**
> |Model|AT $\downarrow$|ADR $\uparrow$|ATR $\uparrow$|RR $\uparrow$|
> |-|-|-|-|-|
> |LLaVA-1.5-Vicuna-7B|    1.00 |	0.62|	0.46      |0.12	|
> |LLaVA-1.5-Vicuna-7B-FIRE100K      |  **0.94**  |  **0.80**  | **0.61** | **0.20** |
>
>
> > *Q3: Why did you use different criteria for filtering dialogues in FIRE-100K and FIRE-1M? For example:*
> > *FIRE-100K: Filtering dialogues with no score improvements or more than 6 turns.
> FIRE-1M: Filtering based on feedback ≥ 8 or the number of turns exceeding 3.*
>
> **A:** FIRE-100K and FIRE-1M employ different data generation methods, leading to distinct data characteristics. These differences prompted us to adopt different data filtering strategies.
>
> For FIRE-100K, we employed a one-pass generation strategy where GPT-4V was provided with both the question and the ground truth to generate the response-feedback dialogue in a single forward pass. Since GPT-4V had access to the ground truth, it typically produced final responses with a score of 8 or higher.
>
> For FIRE-1M, we simulated the response-feedback interaction between a Student model (FIRE-LLaVA) and a Teacher model (FIRE-LLaVA-FD). Unlike GPT-4V, the Student model did not have access to the ground truth, which meant it could not always ensure the accuracy of the final response. Therefore, we filtered the data to retain only those samples with a final response score of 8 or higher. Additionally, based on our observation in FIRE-100K that most dialogues converged within three turns (see Figure 5(d)), we limited the dialogue length in FIRE-1M to three turns to maintain the quality and relevance of the data.
>
> All dialogues without score improvements have been removed from both FIRE-100K and FIRE-1M.
>
> > *Q4: T_baseline Value: What is the value of T_baseline? Since the maximum number of turns is set as n_max = 5, AT should be lower than 1 if T_baseline ≥ 5.*
> Here’s a revised version of your response for improved clarity and precision:
>
> **A:** The value of T_baseline represents the average number of turns taken by the baseline model to reach a final response. For LLaVA-Next-8B, T_baseline is 4.77. We use T_baseline as a normalization factor to scale the absolute number of turns $AT_{abs}$ used in feedback-refinement into a relative metric $AT_
> {rel}$ within the range $[0, 1]$. This normalization, defined as $AT_{rel} = AT_{abs}/T\_baseline$, allows for more meaningful comparisons between different models by accounting for variations in turn usage.
>
>
> > *Q5: In the paper, you mention that if n_k = 1, it is removed from the K samples to compute AT, ADR, ATR, and RR. This implies that K might differ across models. Providing the proportion of n_k = 1 per model would be helpful.*
>
> **A:** The proportion of samples with $n_k=1$ (i.e., those removed from the calculation of AT, ADR, ATR, and RR) can be found in Figure A3 of the Supplementary Material, which shows the accuracy of the first turn. For clarity, we have reorganized these results in the following Table C. Table C shows that the proportion of filtered samples is similar across the three models, ensuring a fair comparison between the models.
>
> **Table C: Proportion of samples with $n_k$ = 1 on FIRE-Bench.**
> |Model|Proportion of removed samples|
> |-|-|
> |LLaVA-Next-8B|46.57%|
> |FIRE-LLaVA-100K|46.77%|
> |FIRE-LLaVA|49.60%|

---

> > ### Author Rebuttal · Authors · 2024-08-22
> >
> > # Rebuttal [2/2]
> > ### Clarity:
> > > Q6: It appears there are 26 datasets used for the FIRE benchmark: 18 for FIRE-100K/FIRE-1M and 8 for unseen datasets in FIRE-Bench.
> >
> > **A:** There are actually **27** data sources used, with **19** contributing to FIRE-100K + FIRE-1M, **8** serving as seen datasets in FIRE-Bench, and another **8** as unseen datasets in FIRE-Bench. The detailed breakdown of these data sources can be found in Section F, Table A4 of the Supplementary Material.
> >
> >
> > **TABLE A4: 27 source datasets for training and test in FIRE.**
> > ||||||
> > | --- | --- | --- | --- | --- |
> > | LLaVA (train) | COCO (train) | SAM (train) | VG (train) | Web-Celebrity (train) |
> > | Web-Landmark (train) | WikiArt (train)  | ALLaVA-Vflan (train) | OCRVQA (train) | Share-TextVQA (train) |
> > | AI2D (train)
> > | VQAV2 (train&test) | GQA (train&test) | ChartQA (train&test) | DocVQA (train&test) | DVQA (train&test) |
> > | GeoQA+ (train&test) | Synthdog-EN (train&test) | TextVQA (train&test) |
> > | LLaVA-in-the-Wild (test) |  MMMU (test) | MME (test) | MM-Vet (test) | SEED-bench (test) |
> > | MathVerse (test) | MathVista (test) |ScienceQA (test) |
> >
> > > *Q7: In Figure 1, highlighting the feedback-refinement component of GPT-4V could better illustrate the similarity between the FIRE model and GPT-4V.*
> >
> > **A:** Thank you for your suggestion. In the revised version, we will highlight the feedback-refinement component of GPT-4V in Figure 1 to better illustrate the similarity between the FIRE model and GPT-4V.
> >
> > > *Q8: For Figure 2, consistent coloring for each label would enhance readability.*
> >
> > **A:** Thank you for your suggestion. We will re-plot the pie charts in Figure 2 to ensure consistent coloring across all three charts, which will enhance readability.
> >
> > > *Q9: For Figure 3, the instruction in the gray part should be reframed as a question for clarity.*
> >
> > **A:**  Thank you for the suggestion. We will reframe the 'instruction' in the gray part as a 'question' and update the color to orange for consistency.
> >
> > ---
> > We hope our responses address your concerns. Please let us know if you have any further questions!

---

> > > ### Comment · Reviewer_q6Ko · 2024-08-28
> > >
> > > Thank you for your response. Your answer addresses my question. If Tables A, B, and C are added to the revised paper, it would be even more helpful. Additionally, clarifying the dataset information would help remove any confusion (e.g., lines 118 and 128). I have now increased my rating.

---

> > > > ### Author Response · Authors · 2024-08-28
> > > > **Thank you for your review!**
> > > >
> > > > Thank you for raising the score! We will add Tables A, B, and C to the revised version to further improve the paper's readability and coherence. We will also include additional dataset information in Sections 3.2 and 3.3 in the revised version.

---

### Comment · Area_Chair_z1v8 · 2024-08-15

**Summary**

This paper introduces FIRE dataset for feedback-refinement with 1.1M multi-turn conversations derived from 27 source datasets.
It enables Vision Language Models (VLMs) to refine responses based on feedback across diverse tasks.
The dataset is collected in two stages: FIRE-100K, gathered by GPT-4V, and FIRE-1M, produced by a model trained on FIRE-100K.
The authors also propose a benchmark called Fire-Bench of 11K feedback-refinement conversations to comprehensively evaluate VLMs' feedback-refining capability under two evaluation settings.
By fine-tuning LLaVA on FIRE-100K and FIRE-1M, they create the FIRE-LLaVA model, a baseline for feedback-refining on the proposed Fire-Bench, outperforming untrained VLMs by 50%.

**Strengths**

- Arguing the need for feedback-refining beyond the standard instruction-following ability of VLMs. This ability to refine the model's own responses would be important for VLMs to progress towards AGI.
- Proposing the first benchmark to evaluate the feedback-refining capability of VLMs.

**Ratings**
6: Marginally above acceptance threshold

**Correctness**

- Not only quantitative comparisons but also qualitative comparisons seem necessary. Additionally, given that this involves a generative model, it would be beneficial to conduct a human study on the outputs produced by models trained on the proposed dataset.
- Need clarified comparison with related work mentioned in below “Relation To Prior Work” part.

**Presentation Clarity**

- This paper well presents necessary information regarding data generation and the proposed benchmark, and is easy for readers to follow.

**Relation To Prior Work**

- This work is closely related to [1], which focuses on generating feedback for self-refinement in Large Language Models (LLMs). In contrast, FIRE utilizes GPT-4V to generate feedback-refinement data. I believe the authors should include a thorough comparison between their work (FIRE) and the approach presented in [1]. This comparison would provide valuable context and highlight the unique contributions of FIRE in the landscape of feedback-refinement methodologies for AI models.
- What are the advantages of this dataset compared to hallucination feedback and refinement datasets like VolCaNo [2]? Are there any unique features beyond its scale? What performance differences would we see on Fire-Bench if models were trained on these different datasets?

[1] Madaan et al., Self-Refine: Iterative Refinement with Self-Feedback, NeurIPS, 2023.

[2] Lee et al., Volcano: Mitigating Multimodal Hallucination through Self-Feedback Guided Revision, NAACL, 2024.

---

> ### Author Rebuttal · Authors · 2024-08-20
>
> # Rebuttal [1/2]
> Thank you so much for your efforts in finding reviewers for our submission. We really appreciate your dedication, especially for taking the time to review our work by yourself. Your support means a lot to us. We will address your suggestions and concerns below.
>
> > *Q1: Not only quantitative comparisons but also qualitative comparisons seem necessary. Additionally, given that this involves a generative model, it would be beneficial to conduct a human study on the outputs produced by models trained on the proposed dataset.*
>
> **A:** Thank you for this valuable feedback. To evaluate the models qualitatively, we conducted a human study comparing responses from LLaVA-Next-8B and FIRE-LLaVA. The interface can be found in Fig. A of the PDF file. We randomly sampled 100 instances and provided each model with identical initial responses and feedback, asking them to generate refined responses. Three independent human evaluators assessed these responses, without knowing which model generated which response (responses were shuffled to ensure blinding). The evaluation results, detailed in Table A, show that FIRE-LLaVA outperforms LLaVA-Next-8B with a significantly higher preference score (37.67 vs. 24.33), indicating that FIRE-LLaVA’s responses are more aligned with human preferences.
>
> **Table A: Human evaluation for LLaVA-Next-8B and FIRE-LLaVA.**
> | |FIRE-LLaVA is Better|	Tie	|LLaVA-Next-8B is Better|
> |-|-|-|-|
> |**Votes**|  37.67 |38  |	24.33  |
>
> > *Q2: This work is closely related to [1], which focuses on generating feedback for self-refinement in Large Language Models (LLMs). In contrast, FIRE utilizes GPT-4V to generate feedback-refinement data. I believe the authors should include a thorough comparison between their work (FIRE) and the approach presented in [1]. This comparison would provide valuable context and highlight the unique contributions of FIRE in the landscape of feedback-refinement methodologies for AI models.*
>
> > *[1] Madaan et al., Self-Refine: Iterative Refinement with Self-Feedback, NeurIPS, 2023.*
>
> **A:** Thank you for pointing out this important comparison. We acknowledge that our work and [1] share a focus on feedback refinement but differ significantly in both **motivation** and **methodology**.
>
> 1. **Motivation**:
>     - **[1] Madaan et al.**: The work presented in [1] aims at self-refinement for NLP tasks, where LLMs improve their responses by iteratively refining their outputs based on self-generated feedback. The primary goal is to enhance the self-improvement capability of LLMs.
>     - **FIRE**: In contrast, our focus is on the feedback-refinement ability of Vision Language Models (VLMs). We aim to enable VLMs to enhance their responses based on human feedback, facilitating more effective interactions between users and models.
>
> 2. **Methodology**:
>     - **[1] Madaan et al.**: This approach relies on **closed-source** **LLMs** that use self-generated feedback to identify and correct their flaws autonomously. This self-feedback mechanism is specific to the capabilities of LLMs and does not directly apply to VLMs.
>     - **FIRE**: Our methodology involves creating a large-scale multimodal feedback-refinement dataset (1.1 million conversations) using GPT-4V and **fine-tuning open-source** **VLMs** on this dataset. This approach is designed to build and enhance the feedback-refining capabilities of VLMs, which do not possess inherent self-feedback capabilities.
>
> We will include a detailed comparison between our work and [1] in the revised version to provide context and highlight the unique contributions of FIRE in advancing feedback-refinement methodologies for multimodal models.

---

> > ### Author Rebuttal · Authors · 2024-08-22
> >
> > # Rebuttal [2/2]
> >
> > > *Q3: What are the advantages of this dataset compared to hallucination feedback and refinement datasets like VolCaNo [2]? Are there any unique features beyond its scale? What performance differences would we see on Fire-Bench if models were trained on these different datasets?*
> > > [2] Lee et al., Volcano: Mitigating Multimodal Hallucination through Self-Feedback Guided Revision, NAACL, 2024.
> >
> > **A:** Compared to VolCaNo, FIRE offers two distinct advantages:
> >
> > 1. **Higher Data Diversity**:
> >     - **VolCaNo** focuses exclusively on refining visual hallucinations and uses the LLaVA dataset for this purpose.
> >     - **FIRE** incorporates data from 27 diverse source datasets, covering a wide range of tasks beyond visual hallucination, including visual reasoning, OCR, document understanding, math reasoning, and chart reasoning. This broader data scope enhances the model’s ability to refine responses across various types of tasks.
> >
> > 2. **Better Data Quality**:
> >     - **VolCaNo** utilizes GPT-3.5 (text-only) for feedback generation, which limits its ability to incorporate visual context.
> >     - **FIRE** employs GPT-4V, which directly integrates images into the feedback generation process, leading to higher quality and more contextually relevant data.
> >
> > To demonstrate the practical advantages of FIRE, we conducted comparative experiments using FIRE-Bench. We first evaluate the released model checkpoint from VolCaNo ([link](https://huggingface.co/kaist-ai/volcano-7b)) on FIRE-Bench. For fairness, we fine-tuned LLaVA-1.5-Vicuna-7B (the same baseline model used in VolCaNo) on FIRE-100K, which is comparable in scale to VolCaNo (127K), and evaluated it on FIRE-Bench.
> >
> > The results are summarized in Table B. The model checkpoint from VolCaNo performed lower on FIRE-Bench compared to the original LLaVA-1.5-Vicuna-7B. In contrast, LLaVA-1.5-Vicuna-7B fine-tuned on FIRE-100K consistently outperformed the model fine-tuned on VolCaNo across all metrics. This demonstrates that models trained on FIRE-100K handle a broader range of tasks more effectively, particularly in complex scenarios such as chart and mathematical reasoning.
> >
> > **Table B: Results of LLaVA-1.5-Vicuna-7B fine-tuned  VolCaNo or FIRE-100K on FIRE-Bench.**
> > |Model|AT $\downarrow$|ADR $\uparrow$|ATR $\uparrow$|RR $\uparrow$|
> > |-|-|-|-|-|
> > |LLaVA-1.5-Vicuna-7B|    1 |	0.62|	0.46      |0.12	|
> > |LLaVA-1.5-Vicuna-7B-VolCaNo	|  1.07  |  0.17  | 0.13 | 0.04 |
> > |LLaVA-1.5-Vicuna-7B-FIRE100K      |  **0.94**  |  **0.80**  | **0.61** | **0.20** |
> >
> > Note that VolCaNo was published at NAACL in June 2024, which is concurrent with our work. We will include this detailed comparison in the revised version of our manuscript to highlight the unique contributions of FIRE and its advantages over existing datasets.
> >
> > ---
> > We hope our responses address your concerns. Please let us know if you have any further questions!

---

### Decision · Program_Chairs · 2024-09-26

**Decision:**

Accept (Poster)

**Comment:**

The paper receives unanimously positive comments and ratings (the AC reviews it personally as there were only 2 reviews submitted) on the value of feedback-refining capability of VLMs. There are not much of negative comments but unfortunately some of the submitted reviews do not meet the bar of regular NeurIPS D&B Track review quality or not submitted. The AC reviews and receives the feedback from the authors and read all other review comments and recommends to accept the submission to NeurIPS 2024 datasets and benchmarks track for its values for the novelty of the benchmark.